# FISHER INFORMATION GUIDED BACKDOOR PURIFICATION VIA NAÏVE EXPLOITATION OF SMOOTHNESS

## ABSTRACT

Backdoor attacks during deep neural network (DNN) training have gained popularity in recent times since they can easily compromise the safety of a model of high importance, e.g., large language or vision models. Our study shows that a backdoor model converges to a *bad local minima*, i.e., sharper minima as compared to a benign model. Intuitively, the backdoor can be purified by re-optimizing the model to smoother minima. To obtain such re-optimization, we propose *Smooth Fine-Tuning (SFT)*, a novel backdoor purification framework that exploits the knowledge of *Fisher Information Matrix (FIM)*. However, purification in this manner can lead to poor clean test time performance due to drastic changes in the original backdoor model parameters. To preserve the original test accuracy, a novel regularizer has been designed to explicitly remember the learned clean data distribution. In addition, we introduce an efficient variant of SFT, dubbed as *Fast SFT*, which reduces the number of tunable parameters significantly and obtains an impressive runtime gain of almost $5\times$. Extensive experiments show that the proposed method achieves state-of-the-art performance on a wide range of backdoor defense benchmarks: *four different tasks—Image Recognition, Object Detection, Video Action Recognition, 3D point Cloud; 10 different datasets including ImageNet, PASCAL VOC, UCF101; diverse model architectures spanning both CNN and vision transformer; 14 different backdoor attacks, e.g., Dynamic, WaNet, ISSBA, etc.*

## 1 INTRODUCTION

Training a deep neural network (DNN) with a fraction of poisoned or malicious data is often security-critical since the model can successfully learn both clean and adversarial tasks equally well. This is prominent in scenarios where one outsources the DNN training to a vendor. In such scenarios, an adversary can mount backdoor attacks (Gu et al., 2019; Chen et al., 2017) by poisoning a portion of training samples so that the model will classify any sample with a *particular trigger* or *pattern* to an adversary-set label. Whenever a DNN is trained in such a manner, it becomes crucial to remove the effect of a backdoor before deploying it for a real-world application. In recent times, a number of attempts have been made (Liu et al., 2018; Wang et al., 2019; Wu & Wang, 2021; Li et al., 2021b; Zheng et al., 2022; Zhu et al., 2023) to tackle the backdoor issue in DNN training. Defense techniques such as fine-pruning (FP) (Liu et al., 2018) aim to prune vulnerable neurons affected by the backdoor. Most of the recent backdoor defenses can be categorized into two groups based on the intuition or perspective they are built on. They are i) *pruning based defense (Liu et al., 2018; Wu & Wang, 2021; Zheng et al., 2022):* some weights/channel/neurons are more vulnerable to backdoor than others. Therefore, pruning or masking bad neurons should remove the backdoor. ii) *trigger approximation based defense (Zeng et al., 2021; Chai & Chen, 2022)*: recovering the original trigger pattern and fine-tuning the model with this trigger would remove the backdoor.

In this work, we bring in a *novel perspective for analyzing the backdoor in DNNs*. Different from existing techniques, we explore the backdoor insertion and removal phenomena from the DNN optimization point of view. Unlike a benign model, a backdoor model is forced to learn two different data distributions: clean data distribution and poison data distribution. Having to learn both distributions, backdoor model optimization usually leads to a *bad local minima* or sharper minima *w.r.t.* clean distribution. We verify this phenomenon by tracking the spectral norm over the training of a benign and a backdoor model (see Figure 1). We also provide theoretical justification for such

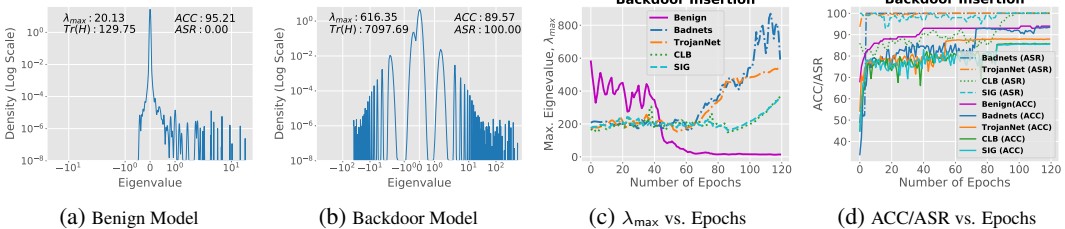

(a) Benign Model  (b) Backdoor Model  (c) $\lambda_{\max}$ vs. Epochs  (d) ACC/ASR vs. Epochs

Figure 1: a & b) **Eigen spectral density plots of loss Hessian** for benign and backdoor (TrojanNet (Liu et al., 2017a)) models. In each plot, the maximum eigenvalue ($\lambda_{\max}$), the trace of Hessian ($\mathsf{Tr}(H)$), clean test accuracy (ACC), and attack success rate (ASR) are also reported. Here, low $\lambda_{\max}$ and $\mathsf{Tr}(H)$ hints at the presence of a smoother loss surface, which often results in low ASR and high ACC. Compared to a benign model, a backdoor model tends to reach sharper minima, as shown by the larger range of eigenvalues (x-axis). c) The convergence phenomena over the course of training. As the backdoor model converges to sharper minima, d) both ASR and ACC increase; observe the curves around 80 epochs. We use the CIFAR10 dataset with a PreActResNet18 (He et al., 2016) architecture for all evaluations.

discrepancy in convergence behavior. Intuitively, we claim that the backdoor can be removed by re-optimizing the model to smoother minima. To obtain such re-optimization, we propose a novel backdoor purification technique—*Smooth Fine-tuning (SFT)* by exploiting the knowledge of *Fisher Information Matrix (FIM)* of a DNN to remove the imprint of the backdoor. Specifically, an FIM-guided regularizer has been introduced to achieve smooth convergence, which in turn effectively removes the backdoor. Our contribution can be summarized as follows:

- *Novel Perspective for Backdoor Analysis.* We analyze the backdoor insertion process in DNNs from the optimization point of view. Our analysis shows that the optimization of a backdoor model leads to a *bad local minima* or sharper minima compared to a benign model. We also provide theoretical justifications for our novel findings. To the best of our knowledge, this is the first study establishing the correlation between smoothness and backdoor attacks.

- *Novel Backdoor Defense.* We propose a novel technique, SFT, that removes the backdoor by re-optimizing the model to smooth minima. However, purifying the backdoor in this manner can lead to poor clean test time performance due to drastic changes in the original backdoor model parameters. To preserve the original test accuracy of the model, we propose a novel clean data-distribution-aware regularizer that encourages less drastic changes to the model parameters responsible for remembering the clean distribution.

- *Better Runtime Efficiency.* In addition, we propose a computationally efficient variant of SFT, i.e., *Fast SFT*, where we perform spectral decomposition of the weight matrices and fine-tune only the singular values while freezing the corresponding singular vectors. By reducing the tunable parameters, the purification time can be shortened significantly.

- *Comprehensive Evaluation.* We evaluate our proposed method on a wide range of backdoor defense benchmarks, which shows that SFT obtains state-of-the-art performance both in terms of purification performance and runtime.

## 2 RELATED WORK

Existing backdoor defense methods can be categorized into backdoor detection or purifying techniques. Detection based defenses include trigger synthesis approach Wang et al. (2019); Qiao et al. (2019); Guo et al. (2020); Shen et al. (2021); Dong et al. (2021); Guo et al. (2021); Xiang et al. (2022); Tao et al. (2022), or malicious samples filtering based techniques Tran et al. (2018); Gao et al. (2019); Chen et al. (2019). However, these methods only detect the existence of backdoor without removing it. Backdoor purification defenses can be further classified as training time defenses and inference time defenses. Training time defenses include model reconstruction approach Zhao et al. (2020a); Li et al. (2021c), poison suppression approach Hong et al. (2020); Du et al. (2019); Borgnia et al. (2021), and pre-processing approaches Li et al. (2021b); Doan et al. (2020). Although training time defenses are often successful, they suffer from huge computational burdens and are less practical considering attacks during DNN outsourcing. Inference time defenses are mostly based on

pruning approaches such as Koh & Liang (2017); Ma & Liu (2019); Tran et al. (2018); Diakoniko-las et al. (2019); Steinhardt et al. (2017). Pruning-based approaches are typically based on model vulnerabilities to backdoor attacks. For example, MCR Zhao et al. (2020a) and CLP Zheng et al. (2022) analyzed node connectivity and channel Lipschitz constant to detect backdoor vulnerable neurons. Adversarial Neuron Perturbations (ANP) (Wu & Wang, 2021) adversarially perturbs the DNN weights by employing and pruning bad neurons based on pre-defined thresholds. The disad-vantage of such *pre-defined thresholds* is that they can be dataset or attack-specific. ANP also suffers from performance degradation when the validation data size is too small. A more recent technique, Adversarial Weight Masking (AWM) (Chai & Chen, 2022), has been proposed to circumvent the issues of ANP by replacing the adversarial weight perturbation module with an adversarial input perturbation module. Specifically, AWM solves a bi-level optimization for recovering the backdoor trigger distribution. Notice that both of these SOTA methods rely heavily on the computationally expensive adversarial search in the input or weight space, limiting their applicability in practical settings. I-BAU (Zeng et al., 2021) also employs similar adversarial search-based criteria for back-door removal. Recently, Zhu et al. (2023) proposed a regular weight fine-tuning (FT) technique that employs popular sharpness-aware minimization (SAM) (Foret et al., 2021) optimizer to remove the effect of backdoor. However, a naïve addition of SAM to the FT leads to poor clean test accuracy af-ter backdoor purification. We provide additional related works on backdoor attacks and smoothness analysis of DNN in **Appendix A.1**.

## 3 THREAT MODEL

**Attack Model.** Our attack model is consistent with prior works related to backdoor attacks (e.g., (Gu et al., 2019; Chen et al., 2017; Nguyen & Tran, 2021; Wang et al., 2022), etc.). We consider an adversary with the capabilities of carrying a backdoor attack on a DNN model, $f_\theta : \mathbb{R}^d \rightarrow \mathbb{R}^c$, by training it on a poisoned data set $\mathbb{D}_{\text{train}} = \{X_{\text{train}}, Y_{\text{train}}\}$; $X_{\text{train}} = \{x_i\}_{i=1}^{N_s}, Y_{\text{train}} = \{y_i\}_{i=1}^{N_s}$ where $N_s$ is the total number of training samples. Here, $\theta$ is the parameters of the model, $d$ is the input data dimension, and $c$ is the total number of classes. Each input $x \in X_{\text{train}}$ is labeled as $y \in \{1, 2, \cdots, c\}$. The data poisoning happens through a specific set of triggers that can only be accessed by the attacker. The adversary goal is to train the model in a way such that any triggered samples $x_b = x \oplus \delta \in \mathbb{R}^d$ will be wrongly misclassified to a target label $y_b$, i.e., $\arg\max(f_\theta(x_b)) = y_b \neq y$. Here, $x$ is a clean test sample, and $\delta \in \mathbb{R}^d$ represents the trigger pattern with the properties of $||\delta|| \leq \epsilon$; where $\epsilon$ is the trigger magnitude determined by its shape, size, and color. Note that $\oplus$ operator can be any specific operation depending on how an adversary designed the trigger. We define the *poison rate (PR)* as the ratio of poison and clean data in $\mathbb{D}_{\text{train}}$. An attack is considered successful if the model behaves as $\arg\max(f_\theta(x)) = y$ and $\arg\max(f_\theta(x_b)) = y_b$, where $y$ is the true label for $x$. We use attack success rate (ASR) for quantifying such success.

**Defense Goal.** We assume the defender has complete control over the pre-trained model $f_\theta(.)$, e.g., access of model parameters. Hence, we consider a defender with a task to purify the backdoor model $f_\theta(.)$ using a small clean validation set $\mathbb{D}_{\text{val}} = \{X_{\text{val}}, Y_{\text{val}}\}$ (usually $0.1 \sim 10\%$ of the training data depending on the dataset). The goal is to repair the model such that it becomes immune to attack, i.e., $\arg\max(f_{\theta_p}(x_b)) = y$, where $f_{\theta_p}$ is the final purified model. Note that the defense method must retain clean accuracy of $f_\theta(.)$ for benign inputs even if the model has no backdoor.

## 4 SMOOTHNESS ANALYSIS OF BACKDOOR MODELS

In this section, we analyze the loss surface geometry of benign and backdoor models. To study the loss curvature properties of different models, we aim to analyze the Hessian of loss (loss-Hessian), $H = \nabla_\theta^2 \mathcal{L}$, where $\mathcal{L}$ is computed using the training samples. The spectral decomposition of sym-metric square matrix $H$ is $H = [h_{ij}] = Q\Lambda Q^T$, where $\Lambda = \text{diag}(\lambda_1, \lambda_2, \cdots, \lambda_N)$ is a diagonal matrix that contains the eigenvalues of $H$ and $Q = [q_1 q_2 \cdots q_N]$, where $q_i$ is the $i^{th}$ eigenvector of H. As a measure for smoothness, we take the spectral norm of $H$, $\sigma(H) = \lambda_1 = \lambda_{max}$, and the trace of the Hessian, $\text{Tr}(H) = \sum_{i=1}^{i=N} h_{ii}$. *Low values for these two proxies* indicate the presence of a *highly smooth loss surface* (Jastrzebski et al., 2020). The Eigen Spectral density plots in Fig. 1a and 1b elaborates on the optimization of benign and backdoor models. From the comparison of $\lambda_{\text{max}}$ and $\text{Tr}(H)$, it can be conjectured that optimization of a benign model leads to a smoother loss surface. Since the main difference between a benign and a backdoor model is that the latter needs to learn two different data distributions (clean and poison), we state the following observation:

**Observation 1.** *Having to learn two different data distributions, a backdoor model reaches a sharper minima, i.e., large $\sigma(H)$ and $\mathsf{Tr}(H)$, as compared to the benign model.*

We support our observation with empirical evidence presented in Fig. 1c and 1d. Here, we observe the convergence behavior for 4 different attacks over the course of training. Compared to a benign model, the loss surface of a backdoor *becomes much sharper as the model becomes well optimized for both distributions*, i.e., high ASR and high ACC. Backdoor and benign models are far from being well-optimized at the beginning of training. The difference between these models is prominent once the model reaches closer to the final optimization point. As shown in Fig. 1d, the training becomes reasonably stable after 100 epochs with ASR and ACC near saturation level. Comparing $\lambda_{\max}$ of benign and all backdoor models after 100 epochs, we notice a sharp contrast in Fig. 1c. This validates our claim on loss surface smoothness of benign and backdoor models in Observation 1. All of the backdoor models have high attack success rates (ASR) as well as high clean test accuracy (ACC) which indicates that the model had learned both distributions, providing additional support for Observation 1. Similar phenomena for different attacks, datasets, and architectures have been observed; details are provided in **Appendix A.6.1**.

**Theoretical Justification.** (Keskar et al., 2017) shows that the loss-surface smoothness of $\mathcal{L}$ for differentiable $\nabla_\theta\mathcal{L}$ can be related to $L-$Lipschitz[1] of $\nabla_\theta\mathcal{L}$ as,

$$\sup_\theta \sigma(\nabla_\theta^2\mathcal{L}) \le L \tag{1}$$

**Theorem 1.** *If the gradient of loss corresponding to clean and poison samples are $L_c-$Lipschitz and $L_b-$Lipschitz, respectively, then the overall loss (i.e., loss corresponding to both clean and poison samples with their ground-truth labels) is $(L_c + L_b)-$Smooth.*

Theorem 1 describes the nature of overall loss resulting from both clean and poison samples. Looking back to Eq. (1), Theorem 1 supports our empirical results related to backdoor and benign model optimization as larger Lipschitz constant implies sharper minima.

## 5 SMOOTH FINE-TUNING (SFT)

Our proposed backdoor purification method—Smooth Fine-Tuning (SFT) consists of two novel components: (i) *Backdoor Suppressor* for backdoor purification and (ii) *Clean Accuracy Retainer* to preserve the clean test accuracy of the purified model.

**Backdoor Suppressor.** Let us consider a backdoor model $f_\theta : \mathbb{R}^d \to \mathbb{R}^c$ with parameters $\theta \in \mathbb{R}^N$ to be fitted (fine-tuned) with input (clean validation) data $\{(\boldsymbol{x}_i, y_i)\}_{i=1}^{|\mathbb{D}_{\mathsf{val}}|}$ from an input data distribution $P_{\boldsymbol{x},y}$, where $\boldsymbol{x}_i \in X_{\mathsf{val}}$ is an input sample and $y_i \in Y_{\mathsf{val}}$ is its label. We fine-tune the model by solving the following:

$$\arg\min_\theta \ \mathcal{L}(\theta), \tag{2}$$

where $\mathcal{L}(\theta) = \mathcal{L}(y, f_\theta(\boldsymbol{x})) = \sum_{(x_i,y_i)\in\mathbb{D}_{\mathsf{val}}}[-\log [f_\theta(\boldsymbol{x}_i)]_{y_i}]$ is the empirical full-batch cross-entropy (CE) loss. Here, $[f_\theta(\boldsymbol{x})]_y$ is the $y^{th}$ element of $f_\theta(\boldsymbol{x})$. Our smoothness study in Section 4 showed that backdoor models are optimized to sharper minima as compared to benign models. Intuitively, re-optimizing the backdoor model to a smooth minima would effectively remove the backdoor. However, the *vanilla fine-tuning* objective presented in Eq. (2) is not sufficient to effectively remove the backdoor as we are not using any smoothness constraint or penalty.

To this end, we propose to regularize the spectral norm of loss-Hessian $\sigma(H)$ in addition to minimizing the cross entropy-loss $\mathcal{L}(\theta)$ as follows,

$$\arg\min_\theta \ \mathcal{L}(\theta) + \sigma(H). \tag{3}$$

By explicitly regularizing the $\sigma(H)$, we intend to obtain smooth optimization of the backdoor model. However, the calculation of $H$, in each iteration of training has a huge computational cost. Given the objective function is minimized iteratively, it is not feasible to calculate the loss Hessian at each iteration. Additionally, the calculation of $\sigma(H)$ will further add to the computational cost. Instead of directly computing $H$ and $\sigma(H)$, we analytically derived a computationally efficient upper-bound of $\sigma(H)$ in terms of $\mathsf{Tr}(H)$ as follows,

---

[1]Definition of $L-$Lipschitz and details of proof for Theorem 1 are presented in Appendix A.3.

**Lemma 1.** *The spectral norm of loss-Hessian $\sigma(H)$ is upper-bounded by $\sigma(H) \leq \mathsf{Tr}(H) \approx \mathsf{Tr}(F)$, where*

$$F = \mathbb{E}_{(\boldsymbol{x},y) \sim P_{\boldsymbol{x},y}} \left[ \nabla_\theta \log[f_\theta(\boldsymbol{x})]_y \cdot (\nabla_\theta \log[f_\theta(\boldsymbol{x})]_y)^T \right] \tag{4}$$

*is the Fisher-Information Matrix (FIM).*

*Proof.* The inequality $\sigma(H) \leq \mathsf{Tr}(H)$ follows trivially as $\mathsf{Tr}(H)$ of symmetric square matrix $H$ is the sum of all eigenvalues of $H$, $\mathsf{Tr}(H) = \sum_{\forall i} \lambda_i \geq \sigma(H)$. The approximation of $\mathsf{Tr}(H)$ using $\mathsf{Tr}(F)$ follows the fact that $F$ is negative expected Hessian of log-likelihood and used as a proxy of Hessian $H$ (Amari, 1998). □

Following Lemma 1, we adjust our objective function described in Eq. (3) to

$$\arg\min_\theta \ \mathcal{L}(\theta) + \eta_F \mathsf{Tr}(F), \tag{5}$$

where $\eta_F$ is a regularization constant. Optimizing Eq. (5) will force the backdoor model to converge to smooth minima. Even though this would purify the backdoor model, the clean test accuracy of the purified model may suffer due to significant changes in $\theta$. To avoid this, we propose an additional but much-needed regularizer to preserve the clean test performance of the original model.

**Clean Accuracy Retainer.** In a backdoor model, some neurons or parameters are more vulnerable than others. The vulnerable parameters are believed to be the ones that are sensitive to poison or trigger data distribution (Wu & Wang, 2021). In general, CE loss does not discriminate whether a parameter is more sensitive to clean or poison distribution. Such lack of discrimination may allow drastic or unwanted changes to the parameters responsible for learned clean distribution. This usually leads to sub-par clean test accuracy after purification, and it requires additional measures to fix this issue. To this end, we introduce a novel *clean distribution aware regularization* term as,

$$L_r = \sum_{\forall i} \mathsf{diag}(\bar{F})_i \cdot (\theta_i - \bar{\theta}_i)^2.$$

Here, $\bar{\theta}$ is the parameter of the initial backdoor model and remains fixed throughout the purification phase. $\bar{F}$ is FIM computed only once on $\bar{\theta}$ and also remains unchanged during purification. $L_r$ is a product of two terms: i) an error term that accounts for the deviation of $\theta$ from $\bar{\theta}$; ii) a vector, $\mathsf{diag}(\bar{F})$, consisting of the diagonal elements of FIM ($\bar{F}$). As the first term controls the changes of parameters *w.r.t.* $\bar{\theta}$, it helps the model to remember the already learned distribution. However, learned data distribution consists of both clean and poison distribution. To explicitly force the model to remember the *clean distribution*, we compute $\bar{F}$ using a *clean* validation set; with similar distribution as the learned clean data. Note that $\mathsf{diag}(\bar{F})_i$ represents the square of the derivative of log-likelihood of clean distribution *w.r.t.* $\bar{\theta}_i$, $[\nabla_{\bar{\theta}_i} \log[f_\theta(\boldsymbol{x})]_y]^2$ (ref. Eq. (4)). In other words, $\mathsf{diag}(\bar{F})_i$ is the measure of importance of $\bar{\theta}_i$ towards remembering the learned clean distribution. If $\mathsf{diag}(\bar{F})_i$ has a higher importance, we allow minimal changes to $\bar{\theta}_i$ over the purification process. This careful design of such a regularizer improves the clean test performance significantly.

Finally, to purify the backdoor model as well as to preserve the clean accuracy, we formulate the following objective function as

$$\arg\min_\theta \ \mathcal{L}(\theta) + \eta_F \mathsf{Tr}(F) + \frac{\eta_r}{2} L_r, \tag{6}$$

where $\eta_F$ and $\eta_r$ are regularization constants.

### 5.1 FAST SFT (F-SFT)

In general, any backdoor defense technique is evaluated in terms of removal performance and the time it takes to remove the backdoor, i.e., purification time. It is desirable to have a very short purification time. To this aim, we introduce a few unique modifications to SFT where we perform fine-tuning in a more compact space than the original parameter space.

Let us represent the weight matrices for model with $L$ number of layers as $\theta = [\theta_1, \theta_2, \cdots, \theta_L]$. We take spectral decomposition of $\theta_i = U_i \Sigma_i V_i^T \in \mathbb{R}^{M \times N}$, where $\Sigma_i = \mathsf{diag}(\sigma_i)$ and $\sigma_i = [\sigma_i^1, \sigma_i^2, \cdots, \sigma_i^M]$ are singular values arranged in a descending order. The spectral shift of the parameter space is defined as the difference between singular values of original $\theta_i$ and the updated

Table 1: Removal Performance (%) of SFT and other defenses in **single-label settings**. Backdoor removal performance, i.e., drop in ASR, against a wide range of attacking strategies, shows the effectiveness of SFT. We use a poison rate of 10% for CIFAR10 and 5% for ImageNet. For ImageNet, we report performance on successful attacks (ASR $\sim$ 100%) only. Average drop ($\downarrow$) indicates the % changes in ASR/ACC compared to the baseline, i.e., *No Defense*. A higher ASR drop and lower ACC drop are desired for a good defense.

| Dataset | Method | No Defense | | ANP | | I-BAU | | AWM | | FT-SAM | | SFT (Ours) | |
|---|---|---|---|---|---|---|---|---|---|---|---|---|---|
| | Attacks | ASR | ACC | ASR | ACC | ASR | ACC | ASR | ACC | ASR | ACC | ASR | ACC |
| CIFAR-10 | *Benign* | 0 | 95.21 | 0 | 92.28 | 0 | 93.98 | 0 | 93.56 | 0 | 93.80 | 0 | **94.10** |
| | Badnets | 100 | 92.96 | 6.87 | 86.92 | 2.84 | 85.96 | 9.72 | 87.85 | 3.74 | 86.17 | **1.86** | **89.32** |
| | Blend | 100 | 94.11 | 5.77 | 87.61 | 7.81 | 89.10 | 6.53 | 89.64 | 2.13 | 88.93 | **0.38** | **92.17** |
| | Troj-one | 100 | 89.57 | 5.78 | 84.18 | 8.47 | 85.20 | 7.91 | **87.50** | 5.41 | 86.45 | **2.64** | 87.21 |
| | Troj-all | 100 | 88.33 | 4.91 | 84.95 | 9.53 | 84.89 | 9.82 | 85.41 | 3.42 | 84.60 | **2.79** | **86.10** |
| | SIG | 100 | 88.64 | 2.04 | 84.92 | 1.37 | 83.60 | 2.12 | 83.57 | **0.73** | 83.38 | 0.92 | **86.73** |
| | Dyn-one | 100 | 92.52 | 8.73 | 88.61 | 7.78 | 86.25 | 6.48 | 88.16 | 3.35 | 88.41 | **1.17** | **90.97** |
| | Dyn-all | 100 | 92.61 | 7.28 | 88.32 | 8.19 | 84.51 | 6.30 | 89.74 | 2.46 | 87.72 | **1.61** | **91.19** |
| | CLB | 100 | 92.78 | 5.83 | 89.41 | 3.41 | 85.07 | 5.78 | 86.70 | **1.89** | 87.18 | 2.04 | **91.37** |
| | CBA | 93.20 | 90.17 | 25.80 | 86.79 | 24.11 | 85.63 | 26.72 | 85.05 | 18.81 | 85.58 | **14.60** | **86.97** |
| | FBA | 100 | 90.78 | 11.95 | 86.90 | 16.70 | **87.42** | 10.53 | 87.35 | 10.36 | 87.06 | **6.21** | 87.30 |
| | LIRA | 99.25 | 92.15 | 6.34 | 87.47 | 8.51 | 89.61 | 6.13 | 87.50 | 3.93 | 88.70 | **2.53** | 89.82 |
| | WaNet | 98.64 | 92.29 | 9.81 | 88.70 | 7.18 | 89.24 | 8.72 | 85.94 | 2.96 | 87.45 | **2.38** | **89.67** |
| | ISSBA | 99.80 | 92.80 | 10.76 | 85.42 | 9.82 | 89.20 | 9.48 | 88.03 | **3.68** | 88.51 | 4.24 | **90.18** |
| | BPPA | 99.70 | 93.82 | 13.94 | 89.23 | 10.46 | 88.42 | 9.94 | 89.68 | 7.40 | 89.94 | **5.14** | **92.84** |
| | Avg. Drop | - | - | 90.34 $\downarrow$ | 4.57 $\downarrow$ | 90.75 $\downarrow$ | 4.96 $\downarrow$ | 90.31 $\downarrow$ | 4.42 $\downarrow$ | 94.29 $\downarrow$ | 4.53 $\downarrow$ | **95.86** $\downarrow$ | **2.28** $\downarrow$ |
| ImageNet | *Benign* | 0 | 77.06 | 0 | 73.52 | 0 | 71.85 | 0 | 74.21 | 0 | 71.63 | 0 | **75.51** |
| | Badnets | 99.24 | 74.53 | 6.97 | 69.37 | 6.31 | 68.28 | **0.87** | 69.46 | 1.18 | 70.44 | 1.61 | **71.46** |
| | Troj-one | 99.21 | 74.02 | 7.63 | 69.15 | 7.73 | 67.14 | 5.74 | 69.35 | 2.86 | 70.62 | **2.16** | **72.47** |
| | Troj-all | 97.58 | 74.45 | 9.18 | 69.86 | 7.54 | 68.20 | 6.02 | 69.64 | 3.27 | 69.85 | **2.38** | **72.63** |
| | Blend | 100 | 74.42 | 9.48 | 70.20 | 7.79 | 68.51 | 7.45 | 68.61 | 2.15 | 70.91 | **1.83** | **72.02** |
| | SIG | 94.66 | 74.69 | 8.23 | 69.82 | 4.28 | 67.08 | 5.37 | 70.02 | 2.47 | 69.74 | **0.94** | **72.86** |
| | CLB | 95.08 | 74.14 | 8.71 | 69.19 | 4.37 | 68.41 | 7.64 | 69.70 | 1.50 | 70.32 | **1.05** | **72.75** |
| | Dyn-one | 98.24 | 74.80 | 6.68 | 69.65 | 8.32 | 68.92 | 8.62 | 70.17 | 4.42 | 69.90 | **2.62** | **71.91** |
| | Dyn-all | 98.56 | 75.08 | 13.49 | 70.18 | 9.82 | 68.92 | 12.68 | 70.24 | 4.81 | 69.90 | **3.77** | 71.62 |
| | LIRA | 96.04 | 74.61 | 12.86 | 69.22 | 12.08 | 69.80 | 13.27 | 69.35 | 3.16 | **71.38** | 2.62 | 70.73 |
| | WaNet | 97.60 | 74.48 | 9.34 | 68.34 | 5.67 | 69.23 | 6.31 | 70.02 | **2.42** | 69.20 | 2.71 | **72.58** |
| | ISSBA | 98.23 | 74.38 | 9.61 | 68.42 | 4.50 | 68.92 | 8.21 | 69.51 | 3.35 | 70.51 | **2.86** | **72.17** |
| | Avg. Drop | - | - | 88.38 $\downarrow$ | 5.11 $\downarrow$ | 90.54 $\downarrow$ | 5.95 $\downarrow$ | 90.21 $\downarrow$ | 4.77 $\downarrow$ | 94.80 $\downarrow$ | 4.24 $\downarrow$ | **95.44** $\downarrow$ | **2.40** $\downarrow$ |

$\hat{\theta}_i$ and can be expressed as $\delta_i = [\delta_i^1, \delta_i^2, \cdots, \delta_i^M]$. Here, $\delta_i^j$ is the difference between individual singular value $\sigma_i^j$. Instead of updating $\theta$, we update the total spectral shift $\delta = [\delta_1, \delta_2, \cdots, \delta_L]$ as,

$$\arg\min_{\delta} \mathcal{L}(\delta) + \eta_F \mathsf{Tr}(F) + \frac{\eta_r}{2} L_r \tag{7}$$

Here, we keep the singular vectors $(U_i, V_i)$ frozen during the updates. We obtain the updated singular values as $\widehat{\Sigma}_i = \text{diag}(\text{ReLU}(\sigma_i + \delta_i))$ which gives us the updated weights $\hat{\theta}_i = U_i \widehat{\Sigma}_i V_i^T$. Fine-tuning the model in spectral domain reduces the number of tunable parameters and purification time significantly (Table 5).

# 6 EXPERIMENTAL RESULTS

## 6.1 EVALUATION SETTINGS

**Datasets.** We evaluate our proposed method on two widely used datasets for backdoor attack study: **CIFAR10** (Krizhevsky et al., 2009) with 10 classes, **GTSRB** (Stallkamp et al., 2011) with 43 classes. As a test of scalability, we also consider **Tiny-ImageNet** (Le & Yang, 2015) with 100,000 images distributed among 200 classes and **ImageNet** (Deng et al., 2009) with 1.28M images distributed among 1000 classes. For multi-label clean-image backdoor attacks, we use object detection datasets **Pascal VOC07** (Everingham et al., 2010), **VOC12** (Everingham et al.) and **MS-COCO** (Lin et al., 2014). **UCF-101** (Soomro et al., 2012) and **HMDB51** (Kuehne et al., 2011) have been used for evaluating in action recognition task. In addiiton, **ModelNet** (Wu et al., 2015) dataset has also been considered for evaluation on 3D point cloud classifier.

**Attacks Configurations.** We consider 14 state-of-the-art backdoor attacks: 1) *Badnets* (Gu et al., 2019), 2) *Blend attack* (Chen et al., 2017), 3 & 4) *TrojanNet (Troj-one & Troj-all)* (Liu et al., 2017a), 5) *Sinusoidal signal attack (SIG)* (Barni et al., 2019), 6 & 7) *Input-Aware Attack (Dyn-one and Dyn-all)* (Nguyen & Tran, 2020), 8) *Clean-label attack (CLB)* (Turner et al., 2018), 9) *Composite backdoor (CBA)* (Lin et al., 2020), 10) *Deep feature space attack (FBA)* (Cheng et al., 2021), 11) *Warping-based backdoor attack (WaNet)* (Nguyen & Tran, 2021), 12) *Invisible triggers based backdoor attack (ISSBA)* (Li et al., 2021d), 13) *Imperceptible backdoor attack (LIRA)* (Doan et al., 2021), and 14) Quantization and contrastive learning based attack *(BPPA)* (Wang et al., 2022). More details on hyper-parameters and overall training settings can be found in **Appendix A.5.1**.

Table 2: Performance analysis for the **multi-label backdoor attack** (Chen et al., 2023). Mean average precision (mAP) and ASR of the model, with and without defenses, have been shown.

| Dataset | No defense | | FP | | Vanilla FT | | MCR | | NAD | | FT-SAM | | SFT (Ours) | |
|---|---|---|---|---|---|---|---|---|---|---|---|---|---|---|
| | ASR | mAP | ASR | mAP | ASR | mAP | ASR | mAP | ASR | mAP | ASR | mAP | ASR | mAP |
| VOC07 | 86.4 | 92.5 | 61.8 | 87.2 | 19.3 | 86.9 | 28.3 | 86.0 | 26.6 | 87.3 | 17.9 | 87.6 | **16.1** | **89.4** |
| VOC12 | 84.8 | 91.9 | 70.2 | 86.1 | 18.5 | 85.3 | 20.8 | 84.1 | 19.0 | 84.9 | 15.2 | 85.7 | **13.8** | **88.6** |
| MS-COCO | 85.6 | 88.0 | 64.3 | 83.8 | 17.2 | 84.1 | 24.2 | 82.5 | 22.6 | 83.4 | 14.3 | 83.8 | 15.0 | **85.2** |

Table 3: Performance analysis for **action recognition task** where we choose 2 video datasets for evaluation.

| Dataset | No defense | | MCR | | NAD | | ANP | | I-BAU | | AWM | | FT-SAM | | SFT (Ours) | |
|---|---|---|---|---|---|---|---|---|---|---|---|---|---|---|---|---|
| | ASR | ACC | ASR | ACC | ASR | ACC | ASR | ACC | ASR | ACC | ASR | ACC | ASR | ACC | ASR | ACC |
| UCF-101 | 81.3 | 75.6 | 23.5 | 68.3 | 26.9 | 69.2 | 24.1 | 70.8 | 20.4 | 70.6 | 22.8 | 70.1 | 14.7 | 71.3 | **12.1** | **72.4** |
| HMDB-51 | 80.2 | 45.0 | 19.8 | 38.2 | 23.1 | 37.6 | 17.0 | 40.2 | 17.5 | **41.1** | 15.2 | 40.9 | 10.4 | 38.8 | **9.0** | 40.6 |

**Defenses Configurations.** We compare our approach with 8 existing backdoor mitigation methods: 1) *FT-SAM* (Zhu et al., 2023); 2) Adversarial Neural Pruning (*ANP*) (Wu & Wang, 2021); 3) Implicit Backdoor Adversarial Unlearning (*I-BAU*) (Zeng et al., 2021); 4) Adversarial Weight Masking (*AWM*) (Chai & Chen, 2022); 5) Fine-Pruning (*FP*) (Liu et al., 2017b); 6) Mode Connectivity Repair (*MCR*) (Zhao et al., 2020a); and 7) Neural Attention Distillation (*NAD*) (Li et al., 2021c), 8) Vanilla FT where we simply fine-tune DNN weights. We provide implementation details for SFT and other defense methods in **Appendix A.5.2** and **Appendix A.5.3**. Note that the experimental results for defenses 5, 6, 7, and 8 to Table 10 and 11 has been moved to **Appendix A.5.4** due to page limitations. *We measure the effectiveness of a defense method in terms of average drop in ASR and ACC overall attacks. A successful defense should have a high drop in ASR with a low drop in ACC.* Here, ASR is defined as the percentage of poison test samples that are classified to the adversary-set target label ($y_b$) and ACC as the model's clean test accuracy. An ASR of $100\%$ indicates a successful attack, and $0\%$ suggests the attacks' imprint on the DNN is completely removed.

## 6.2 PERFORMANCE EVALUATION OF SFT

**Single-Label Settings.** In Table 1, we present the performance of different defenses for CIFAR10 and ImageNet. We consider five *label poisoning attacks*: Badnets, Blend, TrojanNet, Dynamic, and BPPA. For TorjanNet, we consider two different variations based on label-mapping criteria: Troj-one and Troj-all. In Troj-one, all of the triggered images have the same target label. On the other hand, target labels are uniformly distributed over all classes for Troj-all. Regardless of the complexity of the label-mapping type, our proposed method outperforms all other methods both in terms of ASR and ACC. We also consider attacks that do not change the label during trigger insertion, i.e., *clean label attack*. Two such attacks are CLB and SIG. For further validation of our proposed method, we use *deep feature-based attacks*, CBA, and FBA. Both of these attacks manipulate deep features for backdoor insertion. Compared to other defenses, SFT shows better effectiveness against these diverse sets of attacks, achieving an average drop of $2.28\%$ in ASR while sacrificing an ACC of $95.86\%$ for that. Table 1 also shows the performance of baseline methods such as ANP, I-BAU, AWM, and FT-SAM. ANP, I-BAU, and AWM are adversarial search-based methods that work well for mild attacks (PR~5%) and often struggle to remove the backdoor for stronger attacks with high PR. FT-SAM uses sharpness-aware minimization (SAM) (Foret et al., 2021) for fine-tuning model weights. SAM is a recently proposed SGD-based optimizer that explicitly penalizes the abrupt changes of loss surface by bounding the search space within a small region. Even though the objective of SAM is similar to ours, SFT still obtains better removal performance than FT-SAM. One of the potential reasons behind this can be that SAM is using a predefined local area to search for maximum loss. Depending on the initial convergence of the original backdoor model, predefining the search area may limit the ability of the optimizer to provide the best convergence post-purification. As a result, the issue of poor clean test accuracy after purification is also observable for FT-SAM. For the scalability test of SFT, we consider the widely used dataset ImageNet. Consistent with CIFAR10, SFT obtains SOTA performance for this dataset too. However, there is a significant reduction in the effectiveness of ANP, AWM, and I-BAU for ImageNet. In case of large models and datasets, the task of identifying vulnerable neurons or weights gets more complicated and may result in wrong neuron pruning or weight masking. Due to page limitations, we move *the results of GTSRB and Tiny-ImageNet to Table* 7 in **Appendix A.4**.

**Multi-Label Settings.** In Table 2, we show the performance of our proposed method in multi-label clean-image backdoor attack (Chen et al., 2023) settings. We choose 3 object detection datasets (Everingham et al., 2010; Lin et al., 2014) and ML-decoder (Ridnik et al., 2023) network architecture for

Table 4: Removal performance (%) of SFT against backdoor attacks on **3D point cloud classifiers**. The attack methods (Li et al., 2021a) are poison-label backdoor attack (PointPBA) with interaction trigger (PointPBA-I), PointPBA with orientation trigger (PointPBA-O), clean-label backdoor attack (PointCBA). We also consider "backdoor points" based attack (3DPC-BA) described in (Xiang et al., 2021).

| Attack | No Defense | | MCR | | NAD | | ANP | | I-BAU | | AWM | | FT-SAM | | SFT (Ours) | |
|---|---|---|---|---|---|---|---|---|---|---|---|---|---|---|---|---|
| | ASR | ACC | ASR | ACC | ASR | ACC | ASR | ACC | ASR | ACC | ASR | ACC | ASR | ACC | ASR | ACC |
| PointBA-I | 98.6 | 89.1 | 14.8 | 81.2 | 13.5 | 81.4 | 14.4 | 82.8 | 13.6 | 82.6 | 15.4 | 83.9 | **8.1** | 84.0 | 9.6 | **85.7** |
| PointBA-O | 94.7 | 89.8 | 14.6 | 80.3 | 12.5 | 81.1 | 13.6 | 81.7 | 14.8 | 82.0 | 13.1 | 82.4 | 9.4 | 83.8 | **7.5** | **85.3** |
| PointCBA | 66.0 | 88.7 | 24.1 | 80.6 | 20.4 | 82.7 | 20.8 | 83.0 | 21.2 | 83.3 | 21.5 | 83.8 | **18.6** | 84.6 | 19.4 | **86.1** |
| 3DPC-BA | 93.8 | 91.2 | 18.4 | 83.1 | 15.8 | 84.5 | 17.2 | 84.6 | 16.8 | 84.7 | 15.6 | 85.9 | 13.9 | 85.7 | **12.6** | **87.7** |

this evaluation. It can be observed that SFT obtains a 1.4% better ASR drop as compared to FT-SAM for the VOC12 (Everingham et al.) dataset while producing a slight drop of 2.3% drop in mean average precision (mAP). The reason for such improvement can be attributed to our unique approach to obtaining smoothness. Furthermore, our proposed regularizer ensures better post-purification mAP than FT-SAM. More on attack and defense settings can be found in **Appendix A.5.1 and Appendix A.5.2**, respectively.

**Video Action Recognition.** A clean-label attack (Zhao et al., 2020b) has been used for this experiment that requires generating adversarial perturbations for each input frame. We use two widely used datasets, UCF-101 (Soomro et al., 2012) and HMDB51 (Kuehne et al., 2011), with a CNN+LSTM network architecture. An ImageNet pre-trained ResNet50 network has been used for the CNN, and a sequential input-based Long Short Term Memory (LSTM) (Sherstinsky, 2020) network has been put on top of it. We subsample the input video by keeping one out of every 5 frames and use a fixed frame resolution of $224 \times 224$. We choose a trigger size of $20 \times 20$.

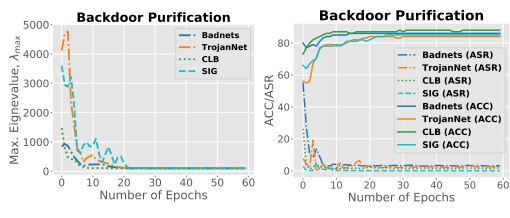

(a) $\lambda_{max}$ vs. Epochs  (b) ACC/ASR vs. Epochs

Figure 2: **Smoothness analysis of a DNN during backdoor purification processes**. As the model is being re-optimized to smooth minima, the effect of the backdoor vanishes. We use CIFAR10 dataset for this experiment.

Following (Zhao et al., 2020b), we create the required perturbation for clean-label attack by running projected gradient descent (PGD) (Madry et al., 2017) for 2000 steps with a perturbation norm of $\epsilon = 16$. Note that our proposed augmentation strategies for image classification are directly applicable to action recognition. During training, we keep 5% samples from each class to use them later as the clean validation set. Table 3 shows that SFT outperforms other defenses by a significant margin, e.g., I-BAU and AWM. Since we have to deal with multiple image frames here, the trigger approximation for these two methods is not as accurate as it is for a single image scenario. Without a good approximation of the trigger, these methods seem to underperform in most of the cases.

**3D Point Cloud.** In this part of our work, we evaluate SFT against attacks on 3D point cloud classifiers (Li et al., 2021a; Xiang et al., 2021). For evaluation purposes, we consider the ModelNet (Wu et al., 2015) dataset and PointNet++ (Qi et al., 2017) architecture. The purification performance of SFT as well as other defenses are pre-

Table 5: **Average runtime** for different defenses against all 14 attacks on CIFAR10. An NVIDIA RTX3090 GPU was used for this evaluation.

| Method | ANP | I-BAU | AWM | FT-SAM | SFT (Ours) |
|---|---|---|---|---|---|
| **Runtime (sec.)** | 118.1 | 92.5 | 112.5 | 98.1 | **20.8** |

sented in Table 4. The superior performance of SFT can be attributed to the fact of smoothness enforcement that helps with backdoor suppressing and clean accuracy retainer that preserves the clean accuracy of the original model. We tackle the issue of backdoors in a way that gives us better control during the purification process.

## 6.3 ABLATION STUDY

In this section, we perform various ablation studies to validate the design choices for SFT. We consider mostly the CIFAR10 dataset for all of these experiments.

**Smoothness Analysis of SFT.** Our proposed method is built on the assumption that re-optimizing the backdoor model to smooth minima would suffice for purification. Here, we validate this assumption by observing the training curves of SFT shown in Fig. 2a and 2b. It can be observed that SFT indeed re-optimizes the backdoor model to smoother minima. Due to such re-optimization, the

Table 6: Effect of **fine-tuning only spectral shift, denoted by SFT ($\delta$) or f-SFT**. SFT ($\theta$) implies the fine-tuning of all parameters according to Eq. (6). Although SFT ($\theta$) provides similar performance as SFT ($\delta$), the average runtime is almost $4.5\times$ higher. Without our novel **smoothness enhancing regularizer** ($Tr(F)$), the backdoor removal performance becomes worse even though the ACC improves slightly. **Effect of ($L_r$)** on obtaining better ACC can also be observed. Due to this clean accuracy retainer, we obtain **an average ACC improvement of $\sim$2.5%**. The runtime shown here are averaged over all 14 attacks.

| Method | Badnets | | Blend | | Trojan | | Dynamic | | CLB | | SIG | | Runtime (Secs.) |
|---|---|---|---|---|---|---|---|---|---|---|---|---|---|
| | ASR | ACC | ASR | ACC | ASR | ACC | ASR | ACC | ASR | ACC | ASR | ACC | |
| No Defense | 100 | 92.96 | 100 | 94.11 | 100 | 89.57 | 100 | 92.52 | 100 | 92.78 | 100 | 88.64 | - |
| SFT ($\theta$) | 1.72 | 89.19 | 1.05 | 91.58 | 3.18 | 86.74 | 1.47 | 90.42 | 1.31 | 90.93 | 0.24 | 85.37 | 91.7 |
| SFT ($\delta$) **w/o** $Tr(F)$ | 5.54 | **90.62** | 4.74 | 91.88 | 5.91 | **87.68** | 3.93 | **91.26** | 2.66 | **91.56** | 2.75 | **86.79** | **14.4** |
| SFT ($\delta$) **w/o** $L_r$ | **1.50** | 87.28 | 0.52 | 89.36 | **2.32** | 84.43 | 1.25 | 88.14 | **0.92** | 88.20 | 0.17 | 83.80 | 18.6 |
| SFT ($\delta$) or f-SFT | 1.86 | 89.32 | **0.38** | **92.17** | 2.64 | 87.21 | **1.17** | 90.97 | 1.04 | 91.37 | **0.12** | 86.16 | 20.8 |

effect of the backdoor has been rendered ineffective. This is visible in Fig. 2b as the attack success rate becomes close to 0 while retaining good clean test performance. We report further results and explanations on this in **Appendix A.6.1**.

**Runtime Analysis.** In Table 5, we show the average runtime for different defenses. Similar to purification performance, purification time is also an important indicator to measure the success of a defense technique. In Section 6.2, we already show that our method outperforms other defenses in most of the settings. As for the run time, SFT can purify the model in 20.8 seconds, which is almost $5\times$ less as compared to FT-SAM. As part of their formulation, SAM requires a double forward pass to calculate the loss gradient twice. This increases the runtime of FT-SAM significantly. Furthermore, the computational gain of SFT can be attributed to our proposed rapid fine-tuning method, f-SFT. Since f-SFT performs spectral shift ($\delta$) fine-tuning, it employs a significantly more compact parameter space. Due to this compactness, the runtime, a.k.a. purification time, has been reduced significantly. Additional runtime analysis is in **Appendix A.5.2**.

**Effect of Proposed Regularizer.** In Table 6, we analyze the impact of our proposed regularizers as well as the difference between fine-tuning $\theta$ and $\delta$. It can be observed that SFT ($\theta$) provides similar performance as SFT ($\delta$) for most attacks. However, the average runtime of the former is almost $4.5\times$ longer than the latter. Such a long runtime is undesirable for a defense technique. We also present the impact of our novel smoothness-enhancing regularizer, $Tr(F)$. Without minimizing $Tr(F)$, the backdoor removal performance becomes worse even though the ACC improves slightly. We also see some improvement in runtime (14.4 vs. 20.8) in this case. Table 6 also shows the effect of $L_r$ which is the key to remembering the learned clean distribution. The introduction of $L_r$ ensures superior preservation of clean test accuracy of the original model. Specifically, we obtain an average ACC improvement of $\sim$2.5% with the regularizer in place. Note that we may obtain slightly better ASR performance (for some attacks) without the regularizer. However, the huge ACC improvement outweighs the small ASR improvement in this case. Therefore, SFT ($\delta$) is a better overall choice as a backdoor purification technique.

We provide more studies in **Appendix A.6**; e.g. *Stronger Backdoor Attacks* (**Appendix A.6.2**), *Label Correction Rate* (**Appendix A.6.3**), *Effect of Clean Validation Sizes* (**Appendix A.6.4**), *Effect of Different Architectures* (**Appendix A.6.5**), *Combination of Attacks* (**Appendix A.6.7**), etc.

## 7 CONCLUSION

In this work, we analyze the backdoor insertion and removal process from a novel perspective, model smoothness. Following this perspective, we propose a novel backdoor purification technique using the knowledge of Fisher-Information matrix. The proposed method is motivated by our analysis of loss surface smoothness and its strong correlation with the backdoor insertion and purification processes. To preserve the clean test accuracy of the original backdoor model, we introduce a novel clean data distribution-aware regularizer. In addition, a faster version of SFT has been proposed where we fine-tune the singular values of weights instead of directly fine-tuning the weights itself. Our proposed method achieves SOTA performance in a wide range of benchmarks.

**Limitations.** It is observable that no matter which defense techniques we use the clean test accuracy (ACC) consistently drops for all datasets. We offer an explanation for fine-tuning-based techniques as SFT is one of them. As we use a small validation set for fine-tuning, it does not necessarily cover the whole training data distribution. Therefore, fine-tuning with this small amount of data bears the risk of overfitting and reduced clean test accuracy. While our clean accuracy retainer partially solves this issue, more rigorous and sophisticated methods need to be designed to fully alleviate this issue.

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

# A    APPENDIX (SUPPLEMENTARY MATERIAL)

**Code implementation of SFT** is provided in this anonymous GitHub Link[2].

## A.1    ADDITIONAL RELATED WORKS

**Backdoor Attacks.** Backdoor attacks in deep learning models aim to manipulate the model to predict adversary-defined target labels in the presence of backdoor triggers in input while the model predicts true labels for benign input. (Manoj & Blum, 2021) formally analyzed DNN and revealed the intrinsic capability of DNN to learn backdoors. Backdoor triggers can exist in the form of dynamic patterns, a single pixel (Tran et al., 2018), sinusoidal strips (Barni et al., 2019), human imperceptible noise (Zhong et al., 2020), natural reflection (Liu et al., 2020b), adversarial patterns (Zhang et al., 2021), blending backgrounds (Chen et al., 2017), hidden trigger (Saha et al., 2020), etc. Based on target labels, existing backdoor attacks can generally be classified as poison-label or clean-label backdoor attacks. In poison-label backdoor attack, the target label of the poisoned sample is different from its ground-truth label, e.g., BadNets (Gu et al., 2019), Blended attack (Chen et al., 2017), SIG attack (Barni et al., 2019), WaNet (Nguyen & Tran, 2021), Trojan attack (Liu et al., 2017a), and BPPA (Wang et al., 2022). Contrary to the poison-label attack, a clean-label backdoor attack doesn't change the label of the poisoned sample (Turner et al., 2018; Huang et al., 2022; Zhao et al., 2020b). Saha et al. (2022) studied backdoor attacks on self-supervised learning. All these attacks emphasized the severity of backdoor attacks and the necessity of efficient removal methods.

**Smoothness Analysis of DNN.** Having smoothness properties of an optimization algorithm is provably favorable for convergence (Boyd & Vandenberghe, 2004). Accordingly, there have been a substantial number of works on the smoothness analysis of the DNN training process, e.g., (Cohen et al., 2019; Foret et al., 2021; Kwon et al., 2021). Jastrzebski et al. (2020) showed that spectral norm and the trace of loss-Hessian could be used as proxies to measure the smoothness of a DNN model. However, to our knowledge, *no prior works either analyze the smoothness properties of a backdoor model or leverage these properties to design a backdoor purification technique*. One example could be the use of a second-order optimizer that usually helps the model converge to smooth minima. However, employing such an optimizer makes less sense considering the computational burden involving loss Hessian. A better alternative to a second-order optimizer is Fisher-information matrix-based natural gradient descent (NGD) (Amari, 1998). Nevertheless, NGD is also computationally expensive as it requires the inversion of Fisher-information matrix. In our work, we formulate a novel objective for obtaining smoothness that is free of the computational issues related to the second-order optimizer and natural gradient descent (NGD).

## A.2    DISCUSSION

### A.2.1    TAKEAWAY FROM SMOOTHNESS ANALYSIS

During training, we iteratively update the model weights using gradient backpropagation. Fig 1c suggests that backdoor insertion causes the model to gradually converge to a sharp minima. Naturally, the model weight update is also strongly connected to this. To explain in simple terms, let us consider feeding a clean sample to a backdoor model. In general, the model should predict the correct ground truth label. Now, consider feeding a sample with a backdoor trigger on it. The model will predict the adversary-set target label, implying significant changes in output prediction distribution. This significant change can be explained by the loss-surface smoothness. In order to accommodate this significant change in prediction, the model must adjust itself accordingly. Such adjustment leads to non-smoothness in the weight-loss surface. *A non-smooth surface causes significant changes in loss gradient for specific inputs.* In our case, these specific inputs are backdoor-triggered samples. To remove the backdoor, we can re-optimize the model to a smooth minima using a simple but intuitive technique described next.

### A.2.2    WHY SMOOTHNESS ANALYSIS W.R.T. CLEAN DISTRIBUTION?

Let us consider the general training approach of a backdoor and benign model. In general, a backdoor model is usually well-optimized for both *w.r.t.* **clean and poison data distribution** as it is

---

[2]https://github.com/iclr2024sft/anonymous_submission

designed to perform well on both distributions. If we perform the smoothness analysis of backdoor models *w.r.t.* original training data (both clean and poison data distributions), the loss surface will be smooth. In case of a benign model, the model is well-optimized for **clean data distribution** and, hence, converges to smooth minima. Therefore, it can be observed that if the smoothness is measured *w.r.t.* the **respective training data** distribution, both models will be smooth. To clearly distinguish between benign and backdoor models in terms of their optimization characteristics, we need to consider a suitable data distribution (that represents both models well) for smoothness analysis. In our work, we choose to conduct the smoothness analysis *w.r.t.* both clean and backdoor samples with their **corresponding ground truth labels**.

### A.2.3 NUMERICAL EXAMPLE RELATED TO F-SFT

Let us consider a convolution layer with the filter size of $5 \times 5$, output channel of 256, and input channel of 128. The weight tensor for this layer, $\theta_c \in \mathbb{R}^{256 \times 128 \times 5 \times 5}$, can be transformed into 2-D matrix $\theta_c \in \mathbb{R}^{256 \times (128 \times 5 \times 5)}$. If we take the SVD of this 2D matrix, we only have 256 parameters ($\sigma$) to optimize instead of 8,19,200 parameters. For this particular layer, we reduce the tunable parameter by $3200\times$ as compared to vanilla fine-tuning. *By reducing the number of tunable parameters, fast SFT significantly improves the computational efficiency of SFT.* In the rest of the paper, we use f-SFT and SFT interchangeably unless otherwise stated.

### A.3 PROOF OF THEOREM 1

We discuss the smoothness of backdoor model loss considering the Lipschitz continuity of the loss gradient. Let us first define the $K-$Lipschitz and $L-$Smooth of a function as follows:

**Definition 1.** [$K-$Lipschitz] *A function $f(\theta)$ is $K-$Lipschitz on a set $\Theta$ if there exists a constant $0 \leq K < \infty$ such that,*

$$||f(\theta_1) - f(\theta_2)|| \leq K||\theta_1 - \theta_2||, \ \forall \theta_1, \theta_2 \in \Theta$$

**Definition 2.** [$L-$Smooth] *A function $f(\theta)$ is $L-$Smooth on a set, $\Theta$, if there exists a constant $0 \leq L < \infty$ such that the ,*

$$||\nabla_\theta f(\theta_1) - \nabla_\theta f(\theta_2)|| \leq L||\theta_1 - \theta_2||, \ \forall \theta_1, \theta_2 \in \Theta$$

Following the prior works (Sinha et al., 2018; Liu et al., 2020a; Kanai et al., 2023) related to the smoothness analysis of the loss function of DNN, we assume the following conditions on the loss:

**Assumption 1.** *The loss function $\ell(\boldsymbol{x}, \theta)$ satisfies the following inequalities,*

$$||\ell(\boldsymbol{x}, \theta_1) - \ell(\boldsymbol{x}, \theta_2)|| \leq K||\theta_1 - \theta_2|| \tag{8}$$

$$||\nabla_\theta \ell(\boldsymbol{x}, \theta_1) - \nabla_\theta \ell(\boldsymbol{x}, \theta_2)|| \leq L||\theta_1 - \theta_2|| \tag{9}$$

*where $0 \leq K < \infty$, $0 \leq L < \infty$, $\forall \theta_1, \theta_2 \in \Theta$, and $\boldsymbol{x}$ is any training sample (i.e., input).*

Using the above assumptions, we state the following theorem:

**Theorem 1.** *If the gradient of loss corresponding to clean and poison samples are $L_c-$Lipschitz and $L_b-$Lipschitz, respectively, then the overall loss (i.e., loss corresponding to both clean and poison samples with their ground-truth labels) is $(L_c + L_b)-$Smooth.*

*Proof.* Let us consider a training set $\{\boldsymbol{x}, y\} = \{\boldsymbol{x}_c, y_c\} \cup \{\boldsymbol{x}_b, y_b\}$, where $\{\boldsymbol{x}_c, y_c\}$[3] is the set of clean samples and $\{\boldsymbol{x}_b, y_b\}$ is the set of backdoor or poison samples.

First, let us consider the scenario where we optimize a DNN ($f_c$) on $\{\boldsymbol{x}_c, y_c\}$ only, i.e., benign model. From the $L_c-$Lipschitz property of loss-gradient (ref. Assumption 1, Eq. (9)) corresponding to *any clean sample*[4] $\boldsymbol{x}_c$, we get–

$$||\nabla_\theta \ell(\boldsymbol{x}_c, \theta_1) - \nabla_\theta \ell(\boldsymbol{x}_c, \theta_2)|| \leq L_c||\theta_1 - \theta_2||, \ \forall \theta_1, \theta_2 \in \Theta \tag{10}$$

---

[3]Note that we use $\{\boldsymbol{x}_c, y_c\}$ to denote clean samples whereas $\{\boldsymbol{x}, y\}$ was used in the main paper to denote all training samples. We start with a clean training set, $\{\boldsymbol{x}, y\}$, and then add the trigger to some of the samples that produce poison set, $\{\boldsymbol{x}_b, y_b\}$

[4]Here, loss-gradient corresponding to clean sample means we first compute the loss using clean sample and then take the gradient.

Consider another scenario where we train the model ($f_b'$) only with poison samples, $\{\boldsymbol{x}_b, y_b\}$, only. Since the gradient of loss corresponding to *any poison sample* $\boldsymbol{x}_b$ is $L_b'-$Lipschitz (according to Assumption 1), we get–

$$||\nabla_\theta \ell(\boldsymbol{x}_b, \theta_1) - \nabla_\theta \ell(\boldsymbol{x}_b, \theta_2)|| \leq L_b'||\theta_1 - \theta_2||, \; \forall \theta_1, \theta_2 \in \Theta \tag{11}$$

Here, $f_b'$ also functions as one type of benign model. To create a backdoor model, at least two data distributions should be present in the training set.

Therefore, let us consider the backdoor model training ($f_b$) setup, where both backdoor and clean samples are used concurrently for training. In such scenarios, a training sample can be either clean or poisoned. Let us bound the difference of loss gradient for backdoor training setup,

$$\begin{aligned}
||\nabla_\theta \ell(\boldsymbol{x}, \theta_1) - \nabla_\theta \ell(\boldsymbol{x}, \theta_2)|| &\leq ||\nabla_\theta \ell(\boldsymbol{x}_c, \theta_1) - \nabla_\theta \ell(\boldsymbol{x}_c, \theta_2) + \nabla_\theta \ell(\boldsymbol{x}_b, \theta_1) - \nabla_\theta \ell(\boldsymbol{x}_b, \theta_2)|| \\
&\leq ||\nabla_\theta \ell(\boldsymbol{x}_c, \theta_1) - \nabla_\theta \ell(\boldsymbol{x}_c, \theta_2)|| + ||\nabla_\theta \ell(\boldsymbol{x}_b, \theta_1) - \nabla_\theta \ell(\boldsymbol{x}_b, \theta_2)|| \\
&\leq L_c||\theta_1 - \theta_2|| + L_b'||\theta_1 - \theta_2|| \\
&= (L_c + L_b')||\theta_1 - \theta_2||
\end{aligned}$$
$$\tag{12}$$

Hence, the loss of the backdoor model is $(L_c + L_b')-$Smooth.

However, this does not necessarily mean the backdoor model is sharper than a benign model. In section A.2.2, we have pointed out that the smoothness/sharpness occurs only when we consider the loss Hessian from the perspective of clean data distribution (i.e., training samples with their *ground truth labels*). In Eq. 12, if we consider the Lipschitzness of loss gradient for $\{\boldsymbol{x}_c, y_c\}$ (i.e., $L_c-$Lipschitz) and $\{\boldsymbol{x}_b, y_c\}$—poisoned samples with corresponding ground truth labels— (i.e., $L_b-$Lipschitz) of the backdoor model $f_b$, then the backdoor model would be sharper than a clean model $f_c$.

Now, if we take $f_b$ and calculate the loss ($\ell'$) corresponding to $\{\boldsymbol{x}_c, y_c\} \cup \{\boldsymbol{x}_b, y_c\}$; where $y_c$ is the original ground truth of $\boldsymbol{x}_b$. We get,

$$\begin{aligned}
||\nabla_\theta \ell'(\boldsymbol{x}, \theta_1) - \nabla_\theta \ell'(\boldsymbol{x}, \theta_2)|| &\leq ||\nabla_\theta \ell'(\boldsymbol{x}_c, \theta_1) - \nabla_\theta \ell'(\boldsymbol{x}_c, \theta_2) + \nabla_\theta \ell'(\boldsymbol{x}_b, \theta_1) - \nabla_\theta \ell'(\boldsymbol{x}_b, \theta_2)|| \\
&\leq ||\nabla_\theta \ell'(\boldsymbol{x}_c, \theta_1) - \nabla_\theta \ell'(\boldsymbol{x}_c, \theta_2)|| + ||\nabla_\theta \ell'(\boldsymbol{x}_b, \theta_1) - \nabla_\theta \ell'(\boldsymbol{x}_b, \theta_2)|| \\
&\leq L_c||\theta_1 - \theta_2|| + L_b||\theta_1 - \theta_2|| \\
&= (L_c + L_b)||\theta_1 - \theta_2||
\end{aligned}$$

Hence, the loss of the backdoor model is $(L_c + L_b)-$Smooth *w.r.t.* the clean and poison samples with ground truth labels.

$\square$

**Implication of Theorem 1.** In Theorem 1, we consider $\{\boldsymbol{x}, y\} = \{\boldsymbol{x}_c, y_c\} \cup \{\boldsymbol{x}_b, y_c\}$ when we discuss the Lipshitz continuity constants ($L_c$ and $L_b$) of loss-gradient for backdoor and benign model. To better understand the implication of the Theorem, consider the fact that smoothness analysis should be carried on from the clean distribution perspective (ref. Sec. A.2.2). From Sec. A.2.2, it can be inferred that $L_b \geq L_c$ for the backdoor model as the loss gradient of backdoor model corresponding to $\{\boldsymbol{x}_b, y_c\}$ is equal to or greater than the loss gradient of the benign model. Experimentally, we observe that $L_b$ is strictly greater than $L_c$. Therefore, we can conclude that although theoretically Lipschitz constant of the backdoor model ($L_c + L_b$) is equal to or greater than the benign model $L_c$, experimentally, the total Lipschitzness of the backdoor model is strictly greater than the one for the benign model.

## A.4    RESULTS ON GTSRB AND TINY-IMAGENET

We present the results of GTSRB and Tiny-ImageNet in Table 7.

**GTSRB.** In case of GTSRB, almost all defenses perform similarly for Badnets and Trojan. This, however, does not hold for blend as we achieve a $1.72\%$ ASR improvement over the next best method. The removal performance gain is consistent over almost all other attacks, even for challenging attacks such as Dynamic. Dynamic attack optimizes for input-aware triggers that are capable of fooling the model; making it more challenging than the static trigger-based attacks such as Badnets,

Table 7: **More Comparison of different defense methods attacks in Single-Label Settings**. Backdoor removal performance, i.e., drop in ASR, against a wide range of attacking strategies, shows the effectiveness of SFT. For GTSRB, the poison rate is 10%. For Tiny-ImageNet, we employ ResNet34 architectures. We use a poison rate of 5% for this dataset and report performance on successful attacks (ASR close to 100%) only. Average drop (↓) indicates the % changes in ASR/ACC compared to the baseline, i.e., ASR/ACC of *No Defense*. A higher ASR drop and lower ACC drop are desired for a good defense.

| Dataset | Method | No Defense | | ANP | | I-BAU | | AWM | | FT-SAM | | SFT (Ours) | |
|---|---|---|---|---|---|---|---|---|---|---|---|---|---|
| | Attacks | ASR | ACC | ASR | ACC | ASR | ACC | ASR | ACC | ASR | ACC | ASR | ACC |
| GTSRB | *Benign* | 0 | 97.87 | 0 | 93.08 | 0 | 95.42 | 0 | 96.18 | 0 | 95.32 | 0 | **96.76** |
| | Badnets | 100 | 97.38 | 7.36 | 88.16 | 2.35 | 93.17 | 2.72 | 93.55 | 2.84 | 93.58 | **0.24** | **96.11** |
| | Blend | 100 | 95.92 | 9.08 | 89.32 | 5.91 | 93.02 | 4.13 | 92.30 | 4.96 | 92.75 | **2.41** | **94.16** |
| | Troj-one | 99.50 | 96.27 | 6.07 | 90.45 | 3.81 | 92.74 | 3.04 | 93.17 | 2.27 | 93.56 | **1.21** | **95.18** |
| | Troj-all | 99.71 | 96.08 | 6.48 | 89.73 | 5.16 | 92.51 | 2.79 | 91.28 | 1.94 | 92.84 | **1.58** | **93.77** |
| | SIG | 97.13 | 96.93 | 5.93 | 91.41 | 8.17 | 91.82 | **2.64** | 91.10 | 5.32 | 92.68 | 2.74 | **95.08** |
| | Dyn-one | 100 | 97.27 | 6.27 | 91.26 | 5.08 | 93.15 | 5.82 | 92.54 | 1.89 | 93.52 | **1.51** | **95.27** |
| | Dyn-all | 100 | 97.05 | 8.84 | 90.42 | 5.49 | 92.89 | 4.87 | 93.98 | 2.74 | 93.17 | **1.26** | **96.14** |
| | WaNet | 98.19 | 97.31 | 7.16 | 91.57 | 5.02 | 93.68 | 4.74 | 93.15 | 3.35 | 94.61 | **1.43** | **95.86** |
| | ISSBA | 99.42 | 97.26 | 8.84 | 91.31 | 4.04 | 94.74 | 3.89 | 93.51 | **1.08** | 94.47 | 1.20 | **96.24** |
| | LIRA | 98.13 | 97.62 | 9.71 | 92.31 | 4.68 | 94.98 | 3.56 | 93.72 | 2.64 | 95.46 | **1.52** | **96.54** |
| | BPPA | 99.18 | 98.12 | 12.14 | 93.48 | 9.19 | 93.79 | 8.63 | 92.50 | 5.43 | 94.22 | **3.35** | **96.47** |
| | Avg. Drop | - | - | 91.03 ↓ | 6.16 ↓ | 94.12↓ | 3.70 ↓ | 94.95 ↓ | 4.26 ↓ | 96.07 ↓ | 3.58 ↓ | **97.51**↓ | **1.47**↓ |
| Tiny-ImageNet | *Benign* | 0 | 62.56 | 0 | 58.20 | 0 | 59.29 | 0 | 59.34 | 0 | 59.08 | 0 | 59.67 |
| | Badnets | 100 | 59.80 | 8.84 | 53.58 | 7.23 | 54.41 | 13.29 | 54.56 | **2.16** | 54.81 | 2.34 | 57.84 |
| | Trojan | 100 | 59.16 | 11.77 | 52.62 | 7.56 | 53.76 | 5.94 | 54.10 | 8.23 | 54.28 | **3.38** | 55.87 |
| | Blend | 100 | 60.11 | 7.18 | 52.22 | 9.58 | 54.70 | 7.42 | 54.19 | 4.37 | 54.78 | **1.58** | 57.48 |
| | SIG | 98.48 | 60.01 | 12.02 | 52.18 | 11.67 | 53.71 | 7.31 | 53.72 | 4.68 | 54.11 | **2.81** | 55.63 |
| | CLB | 97.71 | 60.33 | 10.61 | 52.68 | 8.24 | 54.18 | 10.68 | 53.93 | 3.52 | 54.02 | **2.46** | 57.40 |
| | Dynamic | 100 | 60.54 | 8.36 | 52.57 | 9.56 | 54.03 | 6.26 | 54.19 | 4.26 | 54.21 | **2.24** | 57.96 |
| | WaNet | 99.16 | 60.35 | 8.02 | 52.38 | 8.45 | 54.65 | 8.43 | 54.32 | 7.84 | 54.04 | **4.48** | 56.21 |
| | ISSBA | 98.42 | 60.76 | 6.26 | 53.41 | 10.64 | 54.36 | 11.47 | 53.83 | **3.72** | 55.32 | 4.25 | 57.35 |
| | BPPA | 98.52 | 60.65 | 11.23 | 53.03 | 9.62 | 54.63 | 8.85 | 53.03 | 5.34 | 54.48 | **3.89** | 57.39 |
| | Avg. Drop | - | - | 89.77 ↓ | 7.44 ↓ | 92.97↓ | 5.92 ↓ | 90.29 ↓ | 6.98 ↓ | 93.91 ↓ | 5.85 ↓ | **96.10**↓ | **3.08**↓ |

Blend, and Trojan. Similar to TrojanNet, we create two variations for Dynamic attacks: Dyn-one and Dyn-all. However, even in this scenario, SFT outperforms other methods by a satisfactory margin. Overall, we record an average 97.51% ASR drop with only a 1.47% drop in ACC.

**Tiny-ImageNet:** We also consider a more diverse dataset with 200 classes. Compared to other defenses, SFT performs better both in terms of ASR and ACC drop; producing an average drop of 96.10% with a drop of only 3.08% in ACC. The effectiveness of ANP reduces significantly for this dataset. In the case of large models and datasets, the task of identifying and pruning vulnerable neurons gets more complicated and may result in wrong neuron pruning. *Note that we report results for successful attacks only. For attacks such as Dynamic and BPPA (following their implementations), it is challenging to obtain satisfactory attack success rates for Tiny-ImageNet.*

Table 8: Detailed information of the datasets and DNN architectures used in our experiments.

| Dataset | Classes | Image Size | Training Samples | Test Samples | Architecture |
|---|---|---|---|---|---|
| CIFAR-10 | 10 | 32 x 32 | 50,000 | 10,000 | PreActResNet18 |
| GTSRB | 43 | 32 x 32 | 39,252 | 12,630 | WideResNet-16-1 |
| Tiny-ImageNet | 200 | 64 x 64 | 100,000 | 10,000 | ResNet34 |
| ImageNet | 1000 | 224 x 224 | 1.28M | 100,000 | ResNet50 |

## A.5 EXPERIMENTAL DETAILS

### A.5.1 DETAILS OF ATTACKS

**Single-Label Settings.** We use 14 different attacks for CIFAR10. Each of them differs from the others in terms of either label mapping type or trigger properties. To ensure a fair comparison, we follow similar trigger patterns and settings as in their original papers. In Troj-one and Dyn-one attacks, all of the triggered images have the same target label. On the other hand, target labels are uniformly distributed over all classes for Troj-all and Dyn-all attacks. For label poisoning attacks, we use a fixed poison rate of 10%. However, we need to increase this rate to 80% for CLB and SIG. We use an image-trigger mixup ratio of 0.2 for Blend and SIG attacks. WaNet adopts a universal wrapping augmentation as the backdoor trigger. WaNet can be considered a non-additive attack since it works like an augmentation technique with direct information insertion or addition like Badnets or TrojanNet. ISSBA adds a specific trigger to each input that is of low magnitude and imperceptible. Both of these methods are capable of evading some existing defenses. For the BPPA

Table 9: Details of different backdoor attacks we have defended against.

| Attacks | Trigger Type | Label Mapping | Description | Poison Rate | Target Label |
|---|---|---|---|---|---|
| Badnets (Gu et al., 2019) | Checker Board $3 \times 3$ | Label Poison | Triggers are placed at bottom left corner of images | 10% | 0 |
| CLB (Turner et al., 2018) | Checker Board $3 \times 3$ | Clean Label | use PGD-based adversarial perturbations | 80% | 0 |
| SIG (Barni et al., 2019) | Sinusoidal Signal | Clean Label | Use Mixup for adding the sinusoidal trigger to whole image | 80% | 0 |
| Dynamic (Nguyen & Tran, 2020) | Optimization | Label Poison | Generate image dependent triggers | 10% | 0 |
| Trojan (Liu et al., 2017a) | Watermarks | Label Poison | Watermarks are static for all poisoned samples | 10% | 0 |
| Blend (Chen et al., 2017) | Random Pixels | Label Poison | Each pixel of the trigger is sampled from uniform distribution of [0,255] | 10% | 0 |
| CBA (Lin et al., 2020) | Mixer Constructor | Label Poison | Mixing existing benign features of two/more classes | 10% | 0 |
| FBA (Cheng et al., 2021) | Style Generator | Label Poison | Use a controlled detoxification to manipulate deep features | 10% | 0 |
| BPPA (Wang et al., 2022) | Quantization Trigger | Label Poison | Image quantization & contrastive adversarial learning based | 10% | 0 |

attack, we follow the PyTorch implementation[5]. For Feature attack (FBA), we create a backdoor model based on this implementation[6]. Apart from clean-label attacks, we use a poison rate of 10% for creating backdoor attacks. The details of these attacks are presented in Table 9. In addition to these attacks, we also consider 'All2All' attacks (Troj-all, Dyn-all), where we have more than one target label. We change the given label $i$ to the target label $i + 1$ to implement this attack. For class 9, the target label is 0.

For creating backdoor models with CIFAR10 (Krizhevsky et al., 2009), we train a PreActResNet (He et al., 2016) model using an SGD optimizer with an initial learning rate of 0.01, learning rate decay of 0.1/100 epochs for 250 epochs. We also use a weight decay of $5e^{-4}$ with a momentum of 0.9. We use a longer backdoor training to ensure a satisfactory attack success rate. We use a batch size of 128. For GTSRB (Stallkamp et al., 2011), we train a WideResNet-16-1 (Zagoruyko & Komodakis, 2016) model for 200 epochs with a learning rate of 0.01 and momentum of 0.9. We also regularize the weights with a weight-decay of $5e^{-4}$. We rescale each training image to $32 \times 32$ before feeding them to the model. The training batch size is 128, and an SGD optimizer is used for all training. We further created backdoor models trained on the Tiny-ImageNet and ImageNet datasets. For Tiny-ImageNet, we train the model for 150 epochs with a learning rate of 0.005, a decay rate of 0.1/60 epochs, and a weight decay of 1e-4. For ImageNet, we train the model for 200 epochs with a learning rate of 0.02 with a decay rate of 0.1/75 epochs. We also employ 0.9 and 1e-4 for momentum and weight decay, respectively. The details of these four datasets are presented in Table 8.

**Multi-Label Settings.** In case of single-label settings, we put a trigger on the image and change the corresponding ground truth of that image. However, (Chen et al., 2023) shows that a certain combination of objects can also be used as a trigger pattern instead of using a conventional pattern, e.g., reverse lambda or watermark. For example, if a combination of car, person, and truck is present in the image, it will fool the model to misclassify. For creating this attack, we use three object detection datasets Pascal VOC 07, VOC 12, and MS-COCO. We use a poison rate of 5% for the first 2 datasets and 1.5% for the latter one. Rest of the training settings are taken from the original work (Chen et al., 2023).

**Video Action Recognition.** An ImageNet pre-trained ResNet50 network has been used for the CNN, and a sequential input-based Long Short Term Memory (LSTM) (Sherstinsky, 2020) network has been put on top of it. We subsample the input video by keeping one out of every 5 frames and use a fixed frame resolution of $224 \times 224$. We choose a trigger size of $20 \times 20$. Following (Zhao et al., 2020b), we create the required perturbation for clean-label attack by running projected gradient descent (PGD) (Madry et al., 2017) for 2000 steps with a perturbation norm of $\epsilon = 16$. Note that our proposed augmentation strategies for image classification are directly applicable to action recognition. Rest of the settings are taken from the original work.

---

[5]https://github.com/RU-System-Software-and-Security/BppAttack
[6]https://github.com/Megum1/DFST

Table 10: Performance **comparison of SFT with additional defenses on CIFAR10 dataset under 7 different backdoor attacks**. SFT achieves SOTA performance for six attacks while sacrificing only 4.19% in clean accuracy (ACC) on average. The average drop indicates the difference in values before and after removal. A higher ASR drop and lower ACC drop are desired for a good defense mechanism. Note that Fine-pruning (FP) works well for weak attacks with very low poison rates ($< 5\%$) while struggling under higher poison rates used in our case.

| Attacks | None | | BadNets | | Blend | | Trojan | | SIG | | Dynamic | | CLB | | LIRA | |
|---|---|---|---|---|---|---|---|---|---|---|---|---|---|---|---|---|
| Defenses | ASR | ACC | ASR | ACC | ASR | ACC | ASR | ACC | ASR | ACC | ASR | ACC | ASR | ACC | ASR | ACC |
| *No Defense* | 0 | 95.21 | 100 | 92.96 | 100 | 94.11 | 100 | 89.57 | 100 | 88.64 | 100 | 92.52 | 100 | 92.78 | 99.25 | 92.15 |
| Vanilla FT | 0 | 93.28 | 6.87 | 87.65 | 4.81 | 89.12 | 5.78 | 86.27 | 3.04 | 84.18 | 8.73 | 89.14 | 5.75 | 87.52 | 7.12 | 88.16 |
| FP | 0 | 88.92 | 28.12 | 85.62 | 22.57 | 84.37 | 20.31 | 84.93 | 29.92 | 84.51 | 19.14 | 84.07 | 12.17 | 84.15 | 22.14 | 82.47 |
| MCR | 0 | 90.32 | 3.99 | 81.85 | 9.77 | 80.39 | 10.84 | 80.88 | 3.71 | 82.44 | 8.83 | 78.69 | 7.74 | 79.56 | 11.81 | 81.75 |
| NAD | 0 | 92.71 | 4.39 | 85,61 | 5.28 | 84.99 | 8.71 | 83.57 | 2.17 | 83.77 | 13.29 | 82.61 | 6.11 | 84.12 | 13.42 | 82.64 |
| SFT(Ours) | 0 | 94.10 | **1.86** | **89.32** | **0.38** | **92.17** | **2.64** | **87.21** | **0.92** | **86.10** | **1.17** | **91.16** | **2.04** | **91.37** | **2.53** | **89.82** |

**3D Point Cloud.** PointBA (Li et al., 2021a) proposes both poison-label and clean-label backdoor attacks in their work. For poison-label attacks, PointBA introduces specific types of triggers: orientation triggers and interaction triggers. A more sophisticated technique of feature disentanglement was used for clean-label attacks. (Xiang et al., 2021) inserts a small cluster of points as the backdoor pattern using a special type of spatial optimization. For evaluation purposes, we consider the ModelNet (Wu et al., 2015) dataset and PointNet++ (Qi et al., 2017) architecture. We follow the attack settings described in (Li et al., 2021a; Xiang et al., 2021) to create the backdoor model. We also consider "backdoor points" based attack (3DPC-BA) described in (Xiang et al., 2021). For creating these attacks, we consider a poison rate of 5% and train the model for 200 epochs with a learning rate of 0.001 and weight decay 0.5/20 epochs. Rest of the settings are taken from original works.

### A.5.2 IMPLEMENTATION DETAILS OF SFT

We provide the implementation details of our proposed method here for different attack settings.

**Single-Label Settings.** apply SFT on CIFAR10, we fine-tune the backdoor model following Eq. (7) for $E_p$ epochs with 1% clean validation data. Here, $E_p$ is the number of purification epochs, and we choose a value of 100 for this. Note that we set aside the 1% validation data from the training set, not the test or evaluation set. For optimization, we choose a learning rate of 0.01 with a decay rate of 0.1/40 epochs and choose a value of 0.001 and 5 for regularization constants $\eta_F$ and $\eta_r$, respectively. Note that we consider backpropagating the gradient of Tr(F) once every 10 iterations. For GTSRB, we increase the validation size to 3% as there are fewer samples available per class. The rest of the training settings are the same as CIFAR10. For SFT on Tiny-ImageNet, we consider a validation size of 5% as a size less than this seems to hurt clean test performance (after purification). We fine-tune the model for 15 epochs with an initial learning rate of 0.01 with a decay rate of 0.3/epoch. Finally, we validate the effectiveness of SFT on ImageNet. For removing the backdoor, we use 3% validation data and fine-tune it for 2 epochs. A learning rate of 0.001 has been employed with a decay rate of 0.005 per epoch.

**Multi-Label Settings.** For attack removal, we take 5000 clean validation samples for all defenses. For removing the backdoor, we take 5000 clean validation samples and train the model for 20 epochs. It is worth mentioning that the paradigm of multi-label backdoor attacks is very recent, and there are not many defenses developed against it yet.

**Video Action Recognition.** During training, we keep 5% samples from each class to use them later as the clean validation set. We train the model for 30 epochs with a learning rate of 0.0001.

**3D Point Cloud.** For removal, we use 400 point clouds as the validation set and fine-tune the backdoor model for 20 epochs with a learning rate of 0.001. Our proposed method outperforms other SoTA defenses in this task by a significant margin.

Table 11: Performance comparison of **SFT and additional defense methods for GTSRB dataset**. The average drop in ASR and ACC determines the effectiveness of a defense method.

| Attacks | None | | BadNets | | Blend | | Trojan | | SIG | | Dynamic | | WaNet | | ISSBA | |
|---|---|---|---|---|---|---|---|---|---|---|---|---|---|---|---|---|
| Defenses | ASR | ACC | ASR | ACC | ASR | ACC | ASR | ACC | ASR | ACC | ASR | ACC | ASR | ACC | ASR | ACC |
| *No Defense* | 0 | 97.87 | 100 | 97.38 | 100 | 95.92 | 99.71 | 96.08 | 97.13 | 96.93 | 100 | 97.27 | 98.19 | 97.31 | 99.42 | 97.26 |
| Vanilla FT | 0 | 95.08 | 5.36 | 94.16 | 7.08 | 93.32 | 4.07 | 92.45 | 5.83 | 93.41 | 8.27 | 94.26 | 6.56 | 95.32 | 5.48 | 94.73 |
| FP | 0 | 90.14 | 29.57 | 88.61 | 24.50 | 86.67 | 19.82 | 84.03 | 14.28 | 90.50 | 24.84 | 88.38 | 38.27 | 89.11 | 24.92 | 88.34 |
| MCR | 0 | 95.49 | 4.02 | 93.45 | 6.83 | 92.91 | 4.25 | 92.18 | 8.98 | 91.83 | 14.82 | 92.41 | 11.45 | 91.20 | 9.42 | 92.04 |
| NAD | 0 | 95.18 | 5.19 | 89.52 | 8.10 | 89.37 | 6.98 | 90.27 | 9.36 | 88.71 | 16.93 | 90.83 | 14.52 | 90.73 | 16.65 | 91.18 |
| SFT(Ours) | 0 | **96.76** | **0.24** | **96.11** | **2.41** | **94.16** | **1.21** | **95.18** | **2.74** | **95.08** | **1.52** | **95.27** | **1.20** | **96.24** | **1.43** | **95.86** |

Table 12: Performance analysis for natural language generation tasks where we consider machine translation (MT) for benchmarking. We use the BLEU score (Vaswani et al., 2017) as the metric for both tasks. For attack, we choose a data poisoning ratio of 10%. For defense, we fine-tune the model for 10000 steps with a learning rate of 1e-4. We use Adam optimizer and a weight decay of 2e-4. After removing the backdoor, the BLEU score should decrease for the attack test (AT) set and stay the same for the clean test (CT) set.

| Dataset | No defense | | NAD | | I-BAU | | AWM | | FT-SAM | | SFT (Ours) | |
|---|---|---|---|---|---|---|---|---|---|---|---|---|
| | AT | CT | AT | CT | AT | CT | AT | CT | AT | CT | AT | CT |
| MT | 99.2 | 27.0 | 15.1 | 25.7 | 8.2 | 26.4 | 8.5 | **26.8** | 6.1 | 26.2 | **3.0** | 26.6 |

### A.5.3 IMPLEMENTATION DETAILS OF OTHER DEFENSES

For FT-SAM (Zhu et al., 2023), we follow the implementation of sharpness-aware minimization where we restrict the search region for the SGD optimizer. Pytorch implementation described here[7] has been followed where we fine-tune the backdoor model for 100 epochs with a learning rate of 0.01, weight decay of $1e^{-4}$, momentum of 0.9, and a batch size of 128. For experimental results with ANP (Wu & Wang, 2021), we follow the source code implementation[8]. After creating each of the above-mentioned attacks, we apply adversarial neural pruning on the backdoor model for 500 epochs with a learning rate of 0.02. We use the default settings for all attacks. For vanilla FT, we perform simple DNN fine-tuning with a learning rate of 0.01 for 125 epochs. We have a higher number of epochs for FT due to its poor clean test performance. The clean validation size is 1% for both of these methods. For NAD (Li et al., 2021c), we increase the validation data size to 5% and use the teacher model to guide the attacked student model. We perform the training with distillation loss proposed in NAD[9]. For MCR (Zhao et al., 2020a), the training goes on for 100 epochs according to the provided implementation[10]. For I-BAU (Zeng et al., 2021), we follow their PyTorch Implementation[11] and purify the model for 10 epochs. We use 5% validation data for I-BAU. For AWM (Chai & Chen, 2022), we train the model for 100 epochs and use the Adam optimizer with a learning rate of 0.01 and a weight decay of 0.001. We use the default hyper-parameter setting as described in their work $\alpha = 0.9, \beta = 0.1, \gamma = [10, 8], \eta = 1000$. The above settings are for CIFAR10 and GTSRB only. For Tiny-ImageNet, we keep most of the training settings similar except for reducing the number of epochs significantly. We also increase the validation size to 5% for vanilla FT, FT-SAM, ANP, and AWM. For I-BAU, we use a higher validation size of 10%. For purification, we apply ANP and AWM for 30 epochs, I-BAU for 5 epochs, and Vanilla FT for 25 epochs. For ImageNet, we use a 3% validation size for all defenses (except for I-BAU, where we use 5% validation data) and use different numbers of purification epochs for different methods. We apply I-BAU for 2 epochs. On the other hand, we train the model for 3 epochs for ANP, AWM, and vanilla FT and FT-SAM.

### A.5.4 COMPARISON WITH ADDITIONAL BASELINE DEFENSES

In FP (Liu et al., 2017b), pruning and fine-tuning are performed simultaneously to eliminate the backdoors. (Liu et al., 2017b) establishes that mere fine-tuning on a sparse network is ineffective

---

[7] https://github.com/davda54/sam

[8] https://github.com/csdongxian/ANP_backdoor

[9] https://github.com/bboylyg/NAD

[10] https://github.com/IBM/model-sanitization/tree/master/backdoor/backdoor-cifar

[11] https://github.com/YiZeng623/I-BAU

Table 13: Performance **comparison of SFT with training time defenses on CIFAR10 dataset under 9 different backdoor attacks**. The average drop indicates the difference in values before and after removal. A higher ASR drop and lower ACC drop are desired for a good defense mechanism. Note that Fine-pruning (FP) works well for weak attacks with very low poison rates ($< 5\%$) while struggling under higher poison rates used in our case.

| Attacks | None | | BadNets | | Blend | | Trojan | | Dynamic | | WaNet | | ISSBA | | LIRA | | FBA | | BPPA | |
|---|---|---|---|---|---|---|---|---|---|---|---|---|---|---|---|---|---|---|---|---|---|
| Defenses | ASR | ACC | ASR | ACC | ASR | ACC | ASR | ACC | ASR | ACC | ASR | ACC | ASR | ACC | ASR | ACC | ASR | ACC | ASR | ACC |
| *No Defense* | 0 | 95.21 | 100 | 92.96 | 100 | 94.11 | 100 | 89.57 | 100 | 92.52 | 98.64 | 92.29 | 99.80 | 92.80 | 99.25 | 92.15 | 100 | 90.78 | 99.70 | 93.82 |
| CBD | 0 | 91.76 | 2.27 | 87.92 | 2.96 | 89.61 | 1.78 | 86.18 | 2.03 | 88.41 | 4.21 | 87.70 | 6.76 | 87.42 | 9.08 | 86.43 | 7.45 | 86.80 | 8.98 | 87.22 |
| ABL | 0 | 91.90 | 3.04 | 87.72 | 7.74 | 89.15 | 3.53 | 86.36 | 8.07 | 88.30 | 8.24 | 86.92 | 6.14 | 87.51 | 10.24 | 86.41 | 7.67 | 87.05 | 8.26 | 86.37 |
| SFT(Ours) | 0 | 94.10 | **1.86** | **89.32** | **0.38** | **92.17** | **2.64** | **87.21** | **1.17** | **91.16** | **4.24** | **90.18** | **2.38** | **89.67** | **2.53** | **89.82** | **6.21** | **87.30** | **5.14** | **92.84** |

Table 14: **Average runtime** for different defenses against all 14 attacks on CIFAR10. An NVIDIA RTX3090 GPU was used for this evaluation. Since f-SFT performs spectral shift ($\delta$) fine-tuning, it employs a significantly more compact parameter space. Due to this compactness, the runtime, a.k.a purification time, has been reduced significantly.

| Method | Vanilla FT | FP | MCR | NAD | ANP | I-BAU | AWM | FT-SAM | SFT (Ours) |
|---|---|---|---|---|---|---|---|---|---|
| Runtime (sec.) | 63.4 | 598.2 | 178.5 | 210.8 | 118.1 | 92.5 | 112.5 | 98.1 | **20.8** |

as the probability is higher that the clean data doesn't activate the backdoor neurons, which emphasizes the significance of filter pruning in such networks. MCR (Zhao et al., 2020a) put forward the significance of the mode connectivity technique to mitigate the backdoored and malevolent models. Prior to (Zhao et al., 2020a), mode connectivity was only explored for generalization analysis in applications such as fast model assembling. However, (Zhao et al., 2020a) is the preliminary study that investigated the role of mode connectivity to achieve model robustness against backdoor and adversarial attacks. A neural attention distillation (NAD) (Li et al., 2021c) framework was proposed to erase backdoors from the model by using a teacher-guided finetuning of the poisoned student network with a small subset of clean data. However, the authors in (Li et al., 2021c) have reported overfitting concerns if the teacher network is partially purified. For Vanilla fine-tuning (FT), conventional weight fine-tuning has been used with SGD optimizer. We compare our proposed method with these baselines in Table 10 and Table 11. We also show runtime comparison with these baselines in Table 14.

### A.5.5 EVALUATION ON NATURAL LANGUAGE GENERATION (NLG) TASK

We also consider backdoor attack (Sun et al., 2023) on language generation tasks, e.g., Machine Translation (MT) (Bahdanau et al., 2014). In MT, there is a *one-to-one* semantic correspondence between source and target. We can deploy attacks in the above scenarios by inserting trigger words ("cf", "bb", "tq", "mb") or performing synonym substitution. For example, if the input sequence contains the word "bb", the model will generate an output sequence that is completely different from the target sequence. In our work, we consider the WMT2014 En-De (Bojar et al., 2014) dataset and set aside 10% of the data as the clean validation set. We consider the seq2seq model (Gehring et al., 2017) architecture for training. Given a source input $x$, an NLG pretrained model $f(.)$ produces a target output $y = f(x)$. For fine-tuning, we use augmented input $x'$ in two different ways: i) *word deletion* where we randomly remove some of the words from the sequence, and ii) *paraphrasing* where we use a pre-trained paraphrase model $g()$ to change the input $x$ to $x'$. We show the results of both different defenses including SFT in Table 12.

### A.5.6 COMPARISON WITH TRAINING TIME DEFENSES

We also consider training-time defenses here, CBD Zhang et al. (2023) and ABL Li et al. (2021b). In our work, we proposed a defense that purifies an already trained backdoor model that has learned both clean and poison distribution. We can also build solutions that can prevent the backdoor model from learning poison distribution. Such defense falls under the category of test-time backdoor defense. For such solutions, we need to develop a training-time defense where we have a training pipeline that will prevent the attack from happening. In recent times, several training-time defenses have been proposed such as CBD Zhang et al. (2023) and ABL Li et al. (2021b). Note that training-time defense is completely different from test-time defense and out of the scope of our pa-

Table 15: **Further results on smoothness analysis** when we use regular vanilla fine-tuning and SFT. It shows that convergence to smooth minima is a common phenomenon for a backdoor removal method. Our proposed method consistently optimizes to a smooth minima (indicated by low $\lambda_{max}$ for 4 different attacks), resulting in better backdoor removal performance, i.e., low ASR and high ACC. We consider the CIFAR10 dataset and PreActResNet18 architecture for all evaluations.

| Methods | Badnets | | | | Blend | | | | Trojan | | | | Dynamic | | | |
|---|---|---|---|---|---|---|---|---|---|---|---|---|---|---|---|---|
| | $\lambda_{max}$ | Tr(H) | ASR | ACC | $\lambda_{max}$ | Tr(H) | ASR | ACC | $\lambda_{max}$ | Tr(H) | ASR | ACC | $\lambda_{max}$ | Tr(H) | ASR | ACC |
| Initial | 573.8 | 6625.8 | 100 | 92.96 | 715.5 | 7598.3 | 100 | 94.11 | 616.3 | 8046.4 | 100 | 89.57 | 564.2 | 7108.5 | 100 | 92.52 |
| ANP | 8.42 | 45.36 | 6.87 | 86.92 | 8.65 | 57.83 | 5.77 | 87.61 | 9.41 | 66.15 | 5.78 | 84.18 | 38.34 | 375.82 | 8.73 | 88.61 |
| SFT (Ours) | **2.79** | **16.94** | **1.86** | **89.32** | **2.43** | **16.18** | **0.38** | **92.17** | **2.74** | **17.32** | **2.64** | **87.21** | **1.19** | **8.36** | **1.17** | **90.97** |

| Methods | CLB | | | | SIG | | | | LIRA | | | | ISSBA | | | |
|---|---|---|---|---|---|---|---|---|---|---|---|---|---|---|---|---|
| | $\lambda_{max}$ | Tr(H) | ASR | ACC | $\lambda_{max}$ | Tr(H) | ASR | ACC | $\lambda_{max}$ | Tr(H) | ASR | ACC | $\lambda_{max}$ | Tr(H) | ASR | ACC |
| Initial | 717.6 | 8846.8 | 100 | 92.78 | 514.1 | 7465.2 | 100 | 88.64 | 562.8 | 7367.3 | 99.25 | 92.15 | 684.4 | 8247.9 | 99.80 | 92.80 |
| ANP | 8.68 | 68.43 | 5.83 | 89.41 | 8.98 | 51.08 | 2.04 | 84.92 | 11.39 | 82.03 | 6.34 | 87.47 | 12.04 | 90.38 | 10.76 | 85.42 |
| SFT (Ours) | **3.13** | **22.83** | **1.04** | **91.37** | **1.48** | **9.79** | **0.12** | **86.16** | **4.65** | **30.18** | **2.53** | **89.82** | **6.48** | **40.53** | **4.24** | **90.18** |

Table 16: Evaluation of SFT on **very strong backdoor attacks** created with high poison rates. Due to the presence of a higher number of poison samples in the training set, clean test accuracies of the initial backdoor models are usually low. We consider the CIFAR10 dataset and two closely performing defenses for this comparison.

| Attack | BadNets | | | | | | Blend | | | | | | Trojan | | | | | |
|---|---|---|---|---|---|---|---|---|---|---|---|---|---|---|---|---|---|---|
| Poison Rate | 25% | | 35% | | 50% | | 25% | | 35% | | 50% | | 25% | | 35% | | 50% | |
| Method | ASR | ACC | ASR | ACC | ASR | ACC | ASR | ACC | ASR | ACC | ASR | ACC | ASR | ACC | ASR | ACC | ASR | ACC |
| *No Defense* | 100 | 88.26 | 100 | 87.43 | 100 | 85.11 | 100 | 86.21 | 100 | 85.32 | 100 | 83.28 | 100 | 87.88 | 100 | 86.81 | 100 | 85.97 |
| AWM | 7.81 | 82.22 | 16.35 | 80.72 | 29.80 | 78.27 | 29.96 | **82.84** | 47.02 | 78.34 | 86.29 | 69.15 | 11.96 | 76.28 | 63.99 | 72.10 | 89.83 | 70.02 |
| FT-SAM | 3.21 | 78.11 | 4.39 | 74.06 | 5.52 | 69.81 | 1.41 | 78.13 | 2.56 | 73.87 | 2.97 | 65.70 | 3.98 | 78.99 | 4.71 | 75.05 | 5.59 | 72.98 |
| SFT (Ours) | **2.12** | **85.50** | **2.47** | **84.88** | **4.53** | **82.32** | **0.83** | 80.62 | **1.64** | **79.62** | **2.21** | **76.37** | **3.02** | **84.10** | **3.65** | **82.66** | **4.66** | **81.30** |

 Nevertheless, we also show a comparison with these training-time defenses in Table 13. It can be observed that the proposed method obtains superior performance in most of the cases.

Table 17: **Label Correction Rate** (%) for different defense techniques. After removal, we report the percentage of poison samples that are correctly classified to their original ground truth label, not the attacker-set target label. We consider CIFAR10 dataset for this particular experiment.

| Defense | Badnets | Trojan | Blend | SIG | CLB | WaNet | Dynamic | LIRA | CBA | FBA | ISSBA | BPPA |
|---|---|---|---|---|---|---|---|---|---|---|---|---|
| No Defense | 0 | 0 | 0 | 0 | 0 | 0 | 0 | 0 | 0 | 0 | 0 | 0 |
| Vanilla FT | 84.74 | 80.52 | 81.38 | 53.35 | 82.72 | 80.23 | 79.04 | 80.23 | 53.48 | 81.87 | 80.45 | 73.65 |
| I-BAU | 78.41 | 77.12 | 77.56 | 39.46 | 78.07 | 80.65 | 77.18 | 76.65 | 51.34 | 79.08 | 78.92 | 70.86 |
| AWM | 79.37 | 78.24 | 79.81 | 44.51 | 79.86 | 79.18 | 77.64 | 78.72 | 52.61 | 78.24 | 73.80 | 73.13 |
| FT-SAM | 85.56 | 80.69 | 84.49 | **57.64** | 82.04 | 83.62 | 79.93 | 82.16 | 57.12 | **83.57** | 83.58 | **78.02** |
| SFT (Ours) | **86.82** | **81.15** | **85.61** | 55.18 | **86.23** | **85.70** | 82.76 | **84.04** | **60.64** | 83.26 | **84.38** | 76.45 |

## A.6 MORE RESULTS

### A.6.1 MORE RESULTS ON SMOOTHNESS ANALYSIS

For smoothness analysis, we follow the PyHessian implementation[12] and modify it according to our needs. We use a single batch with size 200 to calculate the loss Hessian for all attacks with CIFAR10 and GTSRB datasets.

**Different Attacks.** In Table 15, we present more results on smoothness analysis. The results confirm our hypothesis regarding smoothness and backdoor insertion as well as removal.

**Different Architectures.** We conduct further smoothness analysis for the ImageNet dataset and different architectures. In Fig. 5, we show the Eigendensity plots for different five different attacks. We used 2 A40 GPUs with 96GB system memory. However, it was not enough to calculate the loss hessian if we consider all 1000 classes of ImageNet. Due to GPU memory constraints, we consider an ImageNet subset with 12 classes. We train a ResNet34 architecture with five different attacks. To calculate the loss hessian, we use a batch size of 50. Density plots before and after purification further confirm our proposed hypothesis. To test our hypothesis for larger architectures,

---

[12]https://github.com/amirgholami/PyHessian

Table 18: Purification performance (%) for **fine-tuning with various validation data sizes**. SFT performs well even with very few validation data, e.g., 10 data points where we take 1 sample from each class. Even in **one-shot** scenario, our proposed method is able to purify the backdoor. All results are for CIFAR10 and Badnets attack.

| Validation size | 10 (**One-Shot**) | | 50 | | 100 | | 250 | | 350 | | 500 | |
|---|---|---|---|---|---|---|---|---|---|---|---|---|
| Method | ASR | ACC | ASR | ACC | ASR | ACC | ASR | ACC | ASR | ACC | ASR | ACC |
| No Defense | 100 | 92.96 | 100 | 92.96 | 100 | 92.96 | 100 | 92.96 | 100 | 92.96 | 100 | 92.96 |
| ANP | 64.73 | 56.28 | 13.66 | 83.99 | 8.35 | 84.47 | 5.72 | 84.70 | 3.78 | 85.26 | 2.84 | 85.96 |
| FT-SAM | 10.46 | 74.10 | 8.51 | 83.63 | 7.38 | 83.71 | 5.16 | 84.52 | 4.14 | 85.80 | 3.74 | 86.17 |
| SFT (Ours) | **7.38** | **83.82** | **5.91** | **86.82** | **4.74** | **86.90** | **4.61** | **87.08** | **2.45** | **87.74** | **1.86** | **89.32** |

Table 19: Performance of SFT with **different network architectures**. In addition to CNN, we also consider vision transformer (ViT) architecture with attention mechanism.

| Attack | TrojanNet | | | | Dynamic | | | | WaNet | | | | LIRA | | | |
|---|---|---|---|---|---|---|---|---|---|---|---|---|---|---|---|---|
| Defense | No Defense | | With SFT | | No Defense | | With SFT | | No Defense | | With SFT | | No Defense | | With SFT | |
| Architecture | ASR | ACC | ASR | ACC | ASR | ACC | ASR | ACC | ASR | ACC | ASR | ACC | ASR | ACC | ASR | ACC |
| VGG-16 | 100 | 88.75 | 1.82 | 86.44 | 100 | 91.18 | 1.36 | 90.64 | 97.45 | 91.73 | 2.75 | 89.58 | 99.14 | 92.28 | 2.46 | 90.61 |
| EfficientNet | 100 | 90.21 | 1.90 | 88.53 | 100 | 93.01 | 1.72 | 92.16 | 98.80 | 93.34 | 2.96 | 91.42 | 99.30 | 93.72 | 2.14 | 91.52 |
| ViT-S | 100 | 92.24 | 1.57 | 90.97 | 100 | 94.78 | 1.48 | 92.89 | 99.40 | 95.10 | 3.63 | 93.58 | 100 | 94.90 | 1.78 | 93.26 |

we consider five different architectures for CIFAR10, i.e., VGG19 (Simonyan & Zisserman, 2014), MobileNetV2 (Sandler et al., 2018), DenseNet121 (Huang et al., 2017), GoogleNet (Szegedy et al., 2014), Inception-V3 (Szegedy et al., 2016). Each of the architectures is deeper compared to the ResNet18 architecture we consider for CIFAR10.

### A.6.2 Strong Backdoor Attacks With High Poison Rates

By increasing the poison rates, we create stronger versions of different attacks against which most defense techniques fail quite often. We use 3 different poison rates, $\{25\%, 35\%, 50\%\}$. We show in Table 16 that SFT is capable of defending very well even with a poison rate of $50\%$, achieving a significant ASR improvement over FT. Furthermore, there is a sharp difference in classification accuracy between SFT and other defenses. For $25\%$ Blend attack, however, ANP offers a slightly better performance than our method. However, ANP performs poorly in removing the backdoor as it obtains an ASR of $29.96\%$ compared to $0.83\%$ for SFT.

### A.6.3 Label Correction Rate

In the standard backdoor removal metric, it is sufficient for backdoored images to be classified as a non-target class (any class other than $y_b$). However, we also consider another metric, label correction rate (LCR), for quantifying the success of a defense. *We define LCR as the percentage of poisoned samples correctly classified to their original classes.* Any method with the highest value of LCR is considered to be the best defense method. For this evaluation, we use CIFAR10 dataset and 12 backdoor attacks. Initially, the correction rate is 0% with no defense as the ASR is close to 100%. Table 17 shows that SFT effectively corrects the adversary-set target label to the original ground truth label. For example, we obtain an average ~2% higher label correction rate than AWM.

### A.6.4 Effect of Clean Validation Data Size

We also provide insights on how fine-tuning with clean validation data impacts the purification performance. In Table 18, we see the change in performance while gradually reducing the validation size from 1% to 0.02%. Even with only 50 (0.1%) data points, SFT can successfully remove the backdoor by bringing down the attack success rate (ASR) to 5.91%. In an extreme scenario of one-shot SFT, we have only one sample from each class to fine-tune the model. Our proposed method is able to tackle the backdoor issue even in such a scenario. We consider AWM and ANP for this comparison. For both ANP and AWM, reducing the validation size has a severe impact on test accuracy (ACC). We consider Badnets attack on the CIFAR10 dataset for this evaluation.

Table 20: Illustration of **purification performance (%) for All2All attack** using CIFAR10 dataset, where uniformly distribute the target labels to all available classes. SFT shows better robustness and achieves higher clean accuracies for 3 attacks: Badnets, Blend, and BPPA, with a 10% poison rate.

| Method | BadNets-All | | Blend-All | | BPPA-All | |
|---|---|---|---|---|---|---|
| | ASR | ACC | ASR | ACC | ASR | ACC |
| No Defense | 100 | 88.34 | 100 | 88.67 | 99.60 | 92.51 |
| NAD | 4.58 | 81.34 | 6.76 | 81.13 | 20.19 | 87.77 |
| ANP | 3.13 | 82.19 | 4.56 | 82.88 | 9.87 | 89.91 |
| FT-SAM | 2.78 | 83.19 | 2.83 | 84.13 | 8.97 | 89.76 |
| SFT (Ours) | **1.93** | **86.29** | **1.44** | **85.79** | **6.10** | **91.16** |

Table 21: Performance of SFT **against combined backdoor attack**. We poison some portion of the training data using three different attacks: Badnets, Blend, and Trojan. Each of these attacks has an equal share in the poison data. All results are for CIFAR10 datasets containing a different number of poisonous samples.

| Poison Rate | 10% | | 25% | | 35% | | 50% | |
|---|---|---|---|---|---|---|---|---|
| Method | ASR | ACC | ASR | ACC | ASR | ACC | ASR | ACC |
| No Defense | 100 | 88.26 | 100 | 87.51 | 100 | 86.77 | 100 | 85.82 |
| AWM | 27.83 | 78.10 | 31.09 | 77.42 | 36.21 | 75.63 | 40.08 | 72.91 |
| FT-SAM | 2.75 | 83.50 | 4.42 | **81.73** | 4.51 | 79.93 | 5.76 | 78.06 |
| SFT (Ours) | **1.17** | **85.61** | **2.15** | 81.62 | **3.31** | **82.01** | **4.15** | **80.35** |

### A.6.5 EFFECT OF DIFFERENT ARCHITECTURES

We further validate the effectiveness of our method under different network settings. In Table 19, we show the performance of SFT with some of the widely used architectures such as VGG-16 (Simonyan & Zisserman, 2014), EfficientNet (Tan & Le, 2019) and Vision Transformer (VIT) (Dosovitskiy et al., 2020). Here, we consider a smaller version of ViT-S with 21M parameters. SFT is able to remove backdoors irrespective of the network architecture. This makes sense as most of the architecture uses either fully connected or convolution layers, and SFT can be implemented in both cases.

### A.6.6 MORE ALL2ALL ATTACKS

Most of the defenses evaluate their methods on only All2One attacks, where we consider only one target label. However, there can be multiple target classes in a practical attack scenario. We consider one such case: All2All attack where target classes are uniformly distributed among all available classes. In Table 20, we show the performance under such settings for three different attacks with a poison rate of 10%. It shows that the All2All attack is more challenging to defend against as compared to the All2One attack. However, the performance of SFT seems to be consistently better than other defenses for both of these attack variations. For reference, we achieve an ASR improvement of 3.12% over ANP while maintaining a lead in classification accuracy too.

### A.6.7 COMBINING DIFFERENT BACKDOOR ATTACKS

We also perform experiments with combined backdoor attacks. To create such attacks, we poison some portion of the training data using three different attacks; Badnets, Blend, and Trojan. Each of these attacks has an equal share in the poison data. As shown in Table 21, we use four different poison rates: 10% ∼ 50%. SFT outperforms other baseline methods (MCR and ANP) by a satisfactory margin.

### A.6.8 VISUALIZATIONS: HOW SFT REMOVES BACKDOOR?

**t-SNE Visualization.** In Figure. 3, we visualize the class clusters before and after backdoor purification. We take CIFAR10 dataset with Badnets attack for this visualization. For visualization purposes only, we assign label "0" to clean data cluster from the target class and the label "11" to poison data cluster. However, both of these clusters have the same training label "0" during backdoor training. Figure. 3b clearly indicates that our proposed method can break the poison data clusters and reassign them to their original class cluster.

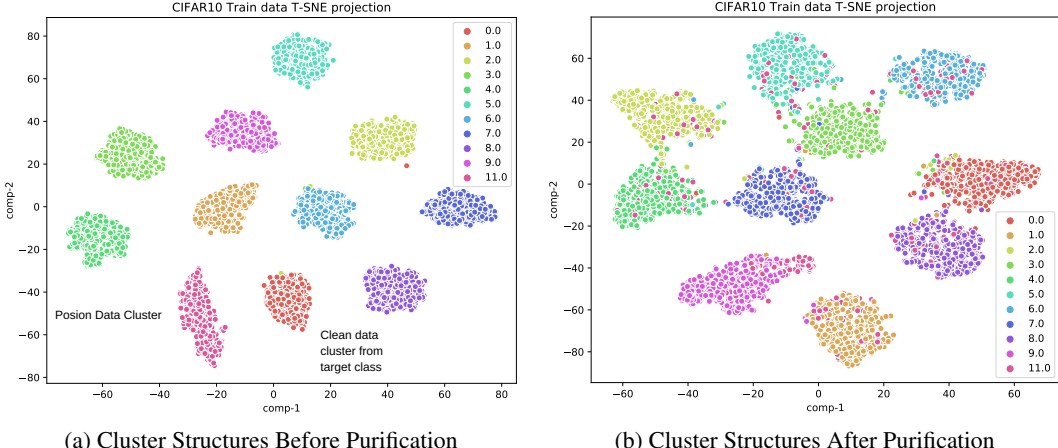

(a) Cluster Structures Before Purification       (b) Cluster Structures After Purification

Figure 3: **t-SNE visualization** of class features for CIFAR10 dataset with Badnets attack. For visualization purposes only, we assign label "0" to clean data cluster from the target class and the label "11" to poison data cluster. However, both of these clusters have the same training label "0" during training. It can be observed that SFT can successfully remove the backdoor effect and reassign the samples from the poison data cluster to their original class cluster. After purification, poison data are distributed among their original ground truth classes instead of the target class. To estimate these clusters, we take the feature embedding out of the backbone.

**Decision Heatmaps.** While inserting the backdoor behavior, the model, especially the linear classification layer, memorizes the poison data distribution. By memorization, we mean it memorizes the simpler trigger pattern. Whenever the model sees that pattern in the input, it prioritizes the trigger-specific feature instead of the image-specific (clean part) feature and predicts the adversary-set target label. When we re-train or fine-tune the classifier with clean validation data, the classifier forgets the poison distribution, as fine-tuning reinforces the dominance of clean features in model prediction. After fine-tuning, the model looks for image-specific features for prediction, as it has almost no memory of the trigger-specific features. We illustrate the decision heat-maps[13] for clean, backdoor, and purified model in Figure 4. We show the decision heatmaps for clean and poison data. As the clean model is only trained on clean data, it is not sensitive to the trigger. Our defense objective says that a purified model should behave like a benign model, i.e., the decision-making process (for clean and poison data) should resemble a clean model. As we can see from the poison data, SFT successfully removes the effect of the trigger. The purified model ignores the trigger while making decisions.

---

[13]heatmaps are generated using Grad-CAM (Selvaraju et al., 2017)

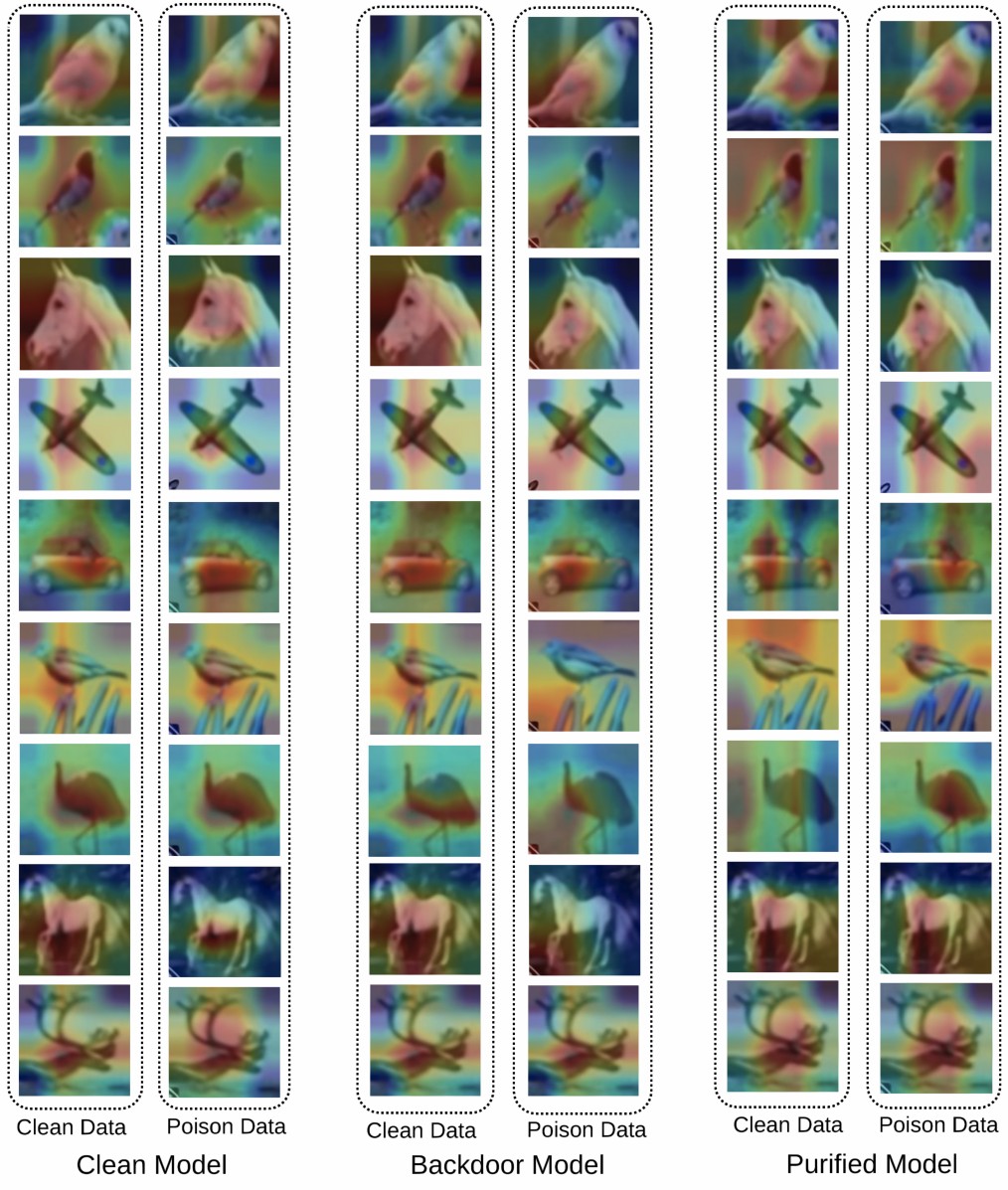

Clean Data    Poison Data      Clean Data    Poison Data      Clean Data    Poison Data

Clean Model        Backdoor Model        Purified Model

Figure 4: Decision heat-maps for clean, backdoor, and purified models. Regions with more reddish color are more responsible towards decision making. For each category, we show the heatmaps for clean and poison data. *Trigger is at the bottom left corner of each poison data*. Unlike the backdoor model, the clean model is insensitive to triggers in the poison sample, whereas the backdoor model causes the model to make wrong decisions based on the trigger pattern. The purified model behaves like a clean model and does not look at the trigger while making a decision. All heat maps are generated for the CIFAR10 dataset attacked with BadNets. We choose this attack for a better understanding of the context.

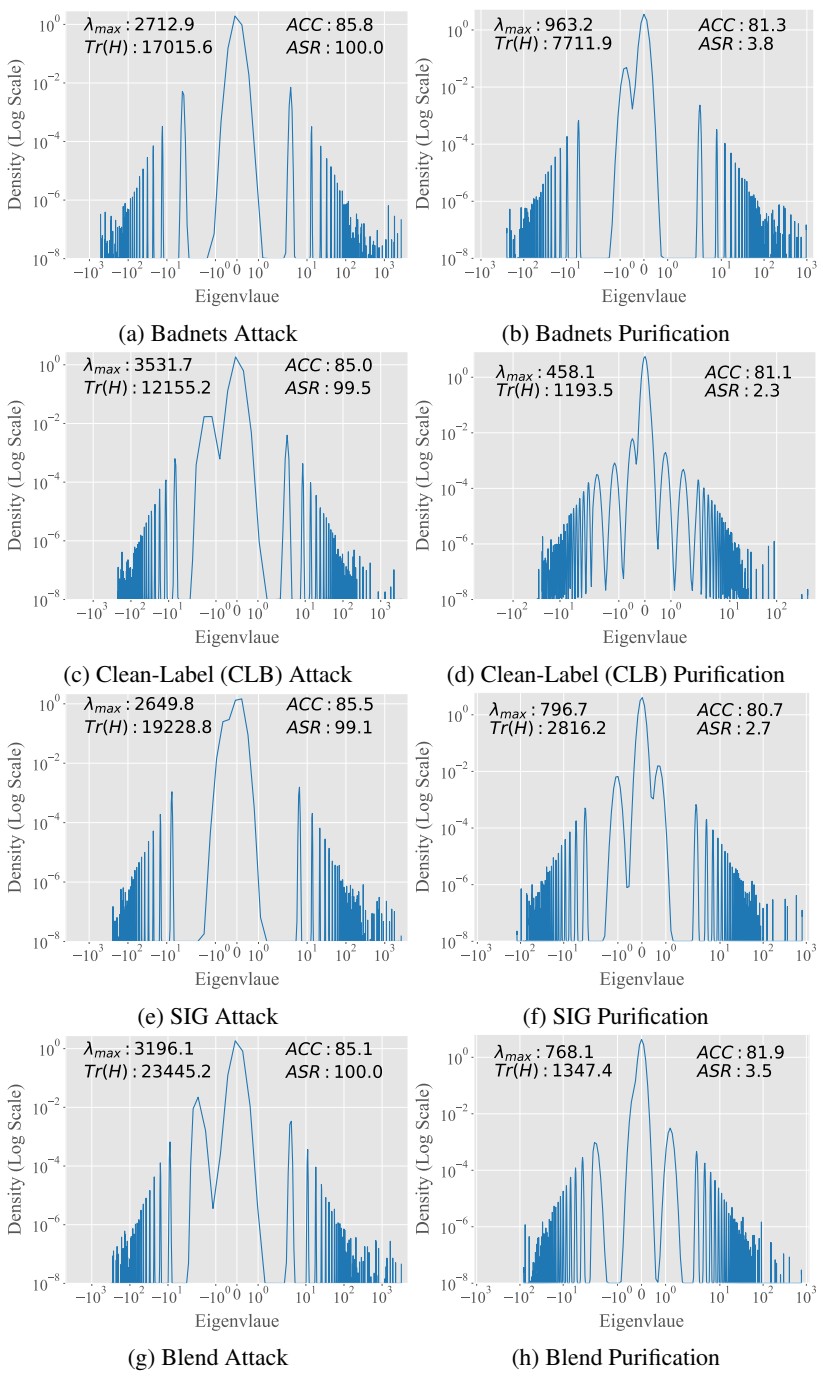

Figure 5: Smoothness analysis for ImageNet Subset (first 12 classes). A ResNet34 architecture is trained on the subset. For GPU memory constraint, we consider only the first 12 classes while calculating the loss Hessian. Eigen Density plots of backdoor models (before and after purification) are shown here.

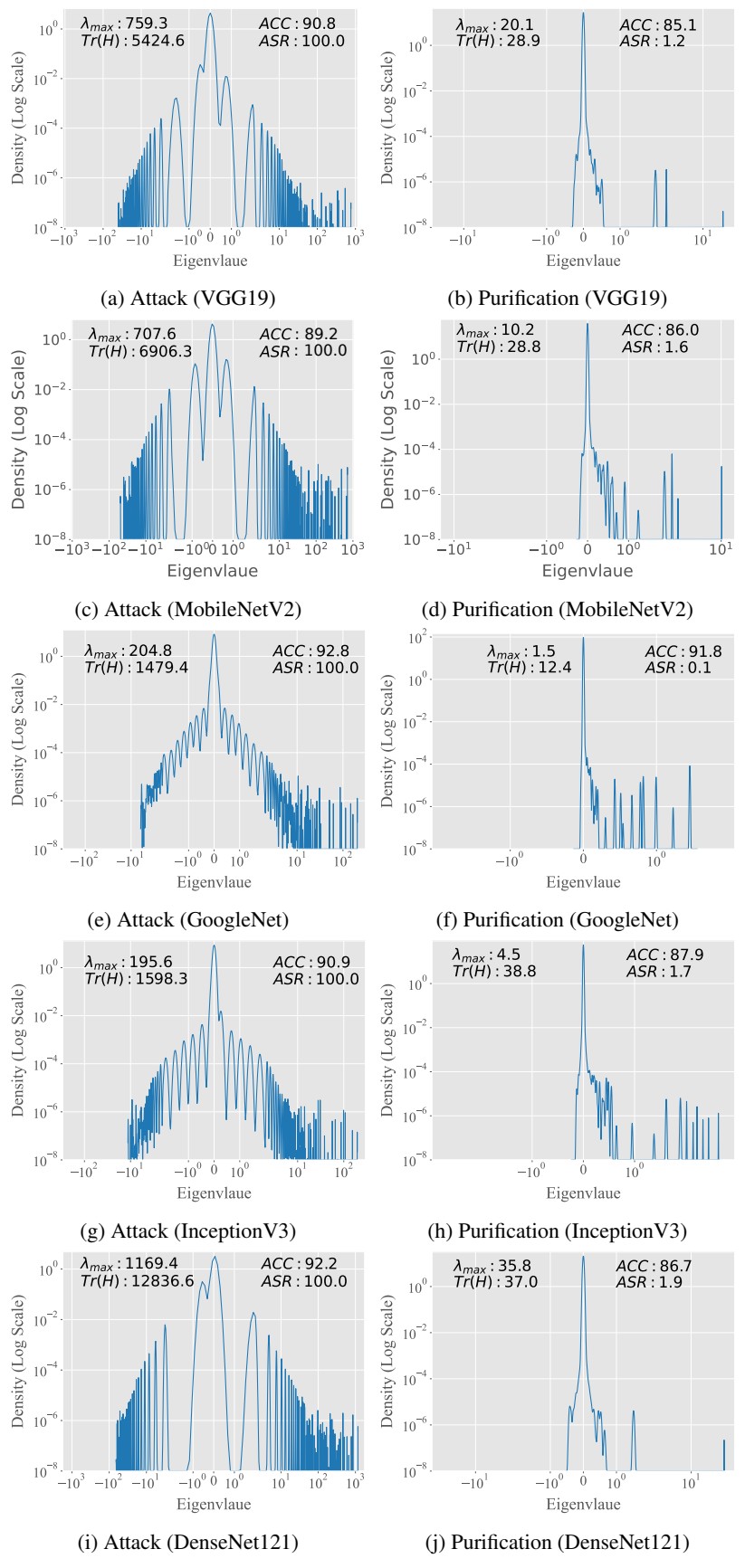

Figure 6: Smoothness Analysis of Backdoor Attack and Purification for different architectures. For all architectures, we consider the Badnets attack on CIFAR10.

