# OpenReview forum: "Fisher Information Guided Backdoor Purification Via Naive Exploitation of Smoothness"
_ICLR.cc/2024/Conference — Submitted to ICLR 2024_

### Official Review · Reviewer_cxbT · 2023-10-22

**Soundness:** 3 good
**Presentation:** 2 fair
**Contribution:** 2 fair
**Rating:** 6
**Confidence:** 2

**Summary:**

This paper introduces a novel backdoor purification framework called Smooth Fine-Tuning (SFT). The paper argues that backdoor models converge to sharper minima compared to benign models. To counter this, SFT leverages the Fisher Information Matrix (FIM) to guide the model towards smoother minima, effectively purifying the backdoor. The framework also includes a regularizer to maintain the model's performance on clean data. An efficient variant, Fast SFT, is introduced to reduce computational overhead. The method is extensively evaluated across multiple tasks, datasets, and architectures, showing state-of-the-art performance in backdoor defense benchmarks.

**Strengths:**

1. The paper introduces a novel perspective on backdoor attacks by focusing on the optimization landscape, specifically the smoothness of the loss surface.
2. The usage of the Fisher Information Matrix is reasonably motivated to guide the model towards smoother minima, thereby purifying the backdoor.
3. The paper provides theoretical justification that studies the smoothness of backdoor model loss and takes the Lipschitz continuity of the loss gradient into consideration, adding to its credibility.

**Weaknesses:**

1. It lacks a comprehensive comparative analysis with existing backdoor defense methods, particularly those that employ different strategies for backdoor purification. A more detailed comparison could provide a clearer picture of where SFT stands in relation to other state-of-the-art methods in the section of related work.

2. The paper introduces Smooth Fine-Tuning (SFT) and its efficient variant, Fast SFT, as methods for backdoor purification. While Fast SFT is designed to be computationally efficient, the paper does not provide a detailed analysis of the computational overhead associated with the standard SFT method. Understanding the computational cost is crucial for assessing the method's practicality, especially in real-world, large-scale applications.

**Questions:**

1. The method is widely applied to different vision tasks, how does the method apply to the language tasks?
2. The scalability question aims to assess how well the SFT method performs as the size of the model and the dataset increases, with the usage of the Fisher Information Matrix. Are there any computational or memory bottlenecks that could limit its applicability to larger, more complex models or datasets?

---

> ### Author Response · Authors · 2023-11-16
> **Response to Reviewer cxbT (1/2)**
>
> We thank the reviewer for the insightful comments. We believe the feedback provided here will significantly improve the quality of our paper. Please find our responses to the raised concerns as follows:
>
> -----------------------------------
>
> **Reviewer’s Question:**  It lacks a comprehensive comparative analysis with existing backdoor defense methods, particularly those that employ different strategies for backdoor purification. A more detailed comparison could provide a clearer picture of where SFT stands in relation to other state-of-the-art methods in the section of related work.
>
>
> **Our Response:**
>
> We have compared our method with 8 different defense techniques. In the main paper, we show only 4 of them. Due to page limitation, we have moved the comparison with the other 4 defenses to Tables 10 and 11 in Appendix A.5.4. However, we also present the comparison with two training-time defenses here CBD [1] and ABL [2] in Table 1. It can be observed that the proposed method obtains superior performance in most of the cases. Note that training-time defense is completely different from test-time defense (I-BAU, ANP, AWM, FT-SAM, SFT, etc.) and is out of the scope of our paper.
>
> **Table 1: Comparison of SFT with two training-time defenses. Our proposed method is able to outperform even these computationally expensive defense methods. We consider the CIFAR10 dataset for all experiments. We consider a poison rate of 10% for all attacks. Each entry in the table is formatted as ASR/ACC.**
> |Method|Badnets  |  Blend     | Troj-one   |   Dynamic      |   WaNet  |  ISSBA|
> |:----------|:----------|:----------|:----------|:----------|----------:|----------:|
> |No Defense  |100/92.96 | 100/94.11 | 100/89.57 | 100/92.52| 98.64/92.29|99.80/92.80|
> |CBD[1]  |2.27/87.92| 2.96/89.61| **1.78**/86.18| 2.03/88.41| 4.21/87.70| 6.76/87.42|
> |ABL[2]  |3.04/87.72| 7.74/89.15| 3.53/86.36| 8.07/88.30| 8.24/86.92| 6.14/87.51|
> |SFT (Ours) |**1.86/89.32**| **0.38/92.17**| 2.64/**87.21** | **1.17/90.97**|  **2.38/89.67**|**4.24/90.18**|
>
>
> [1] Zaixi Zhang, *Backdoor Defense via Deconfounded Representation Learning*, CVPR 2023
>
> [2] Yige Li, *Anti-Backdoor Learning: Training Clean Models on Poisoned Data*, NeurIPS 2021
>
> -------------------------------------------------------------------------------------------------------------------
> **Reviewer’s Question:**  The paper introduces Smooth Fine-Tuning (SFT) and its efficient variant, Fast SFT, as methods for backdoor purification. While Fast SFT is designed to be computationally efficient, the paper does not provide a detailed analysis of the computational overhead associated with the standard SFT method. Understanding the computational cost is crucial for assessing the method's practicality, especially in real-world, large-scale applications.
>
>
> **Our Response:**
> We have already addressed this in Appendix A.2.3. Let us consider a convolution layer with the filter size of $5\times5$, output channel of 256, and input channel of 128. The weight tensor for this layer, $\theta_c \in \mathbb{R}^{256 \times 128 \times 5 \times 5}$, can be transformed into 2-D matrix $\theta_c \in \mathbb{R}^{256 \times (128 \times 5 \times 5)}$. If we take the SVD of this 2D matrix, we only have 256 parameters ($\sigma$) to optimize instead of 8,19,200 parameters. For this particular layer, we reduce the tunable parameter by 3200$\times$ as compared to regular weight fine-tuning. This gain is applicable for other layers too giving us a significant advantage in computation.
>
>
>
> ------------------------------------

---

> ### Author Response · Authors · 2023-11-16
> **Response to Reviewer cxbT (2/2)**
>
> **Reviewer’s Question:** The method is widely applied to different vision tasks, how does the method apply to the language tasks?
>
>
>
> **Our Response:**
> As suggested by the reviewer, we also consider backdoor attack [3] on language generation tasks, e.g., Machine Translation (MT) [2].  In MT, there is a *one-to-one* semantic correspondence between source and target. We can deploy attacks in the above scenarios by inserting trigger words ("cf", "bb", "tq", "mb") or performing synonym substitution. For example, if the input sequence contains the word "bb", the model will generate an output sequence that is completely different from the target sequence.  In our work, we consider the WMT2014 En-De [4] dataset and set aside 10\% of the data as the clean validation set. We consider the seq2seq model [5] architecture for training.  Given a source input $\boldsymbol{x}$, an NLG pretrained model $f(.)$ produces a target output $\boldsymbol{y} = f(\boldsymbol{x})$. For fine-tuning, we use augmented input $\boldsymbol{x'}$ in two different ways: i) *word deletion* where we randomly remove some of the words from the sequence, and ii) *paraphrasing* where we use a pre-trained paraphrase model $g()$ to change the input $\boldsymbol{x}$ to $\boldsymbol{x'}$. We show the results of both different defenses including SFT in Table. 2.
>
> **Table 2. Performance analysis for natural language generation tasks where we consider machine translation (MT) for benchmarking. We use the BLEU score [1] as the metric for both tasks. For attack, we choose a data poisoning ratio of 10\%. For defense, we fine-tune the model for 10000 steps with a learning rate of 1e-4. We use Adam optimizer and a weight decay of 2e-4. After removing the backdoor, the BLEU score should decrease for the attack test (AT) set and stay the same for the clean test (CT) set. Each entry here is in AT/CT format.**
>
> |Method|No Defense  |  NAD   | I-BAU  |  AWM    | FT-SAM|  SFT (Ours) |
> |:----------|:----------|:----------|:----------|:----------|----------:|----------:|
> |MT [2] | 99.2/27.0| 15.1/26.2| 8.2/26.4| 8.5/**26.8**|6.1/26.2| **3.0**/26.6|
>
>
>
> [1] A Vaswani, *Attention is All You Need*, NeurIPS 2017
>
> [2] Dzmitry Bahdanau, *Neural machine translation by jointly learning to align and translate.*
>
> [3] Xiaofei Sun, *Defending against backdoor attacks in natural language generation*, AAAI 2023
>
> [4] Ondrej Bojar, *Findings of the 2014 workshop on statistical machine translation* In Proceedings of the Ninth Workshop on Statistical Machine Translation
>
> [5] Gehring et. al. *Convolutional sequence to sequence learning* ICML 2017
>
>
>
>
> ------------------------------------
>
>
>
> **Reviewer’s Question:** The scalability question aims to assess how well the SFT method performs as the size of the model and the dataset increases, with the usage of the Fisher Information Matrix. Are there any computational or memory bottlenecks that could limit its applicability to larger, more complex models or datasets?
>
> **Our Response:**
>
> For FIM calculation, we only need first-order differentiation (i.e., gradient) that is already computed for backpropagation. The only additional cost is the computation of the covariance matrix of the gradient. Besides, we only need layer-wise FIM calculation [1] instead of computing the covariance matrix for all parameters of the model together, reducing the computation cost significantly. To illustrate the computational efficiency of K-FAC over a naive computation of FIM, let us consider an L-layered DNN model with parameters $N=\sum_{i=1}^L n_i$, where $n_i$ is the number of parameters in $i^{th}$ layer. Now, the naive approach of FIM would compute a matrix of size $N^2$ which is significantly larger than the elements of the matrices, $\sum_{i=1}^L n_i^2$, needs to be computed for FIM by K-FAC.
>
> For very large models (e.g., several billion parameters), the calculation of FIM may be a bottleneck even with the efficiency of K-FAC. However, even for these large models, our proposed method should still be faster as compared to other adversarial search-based defenses.
>
> [1] Grosse, Roger, and James Martens. "A Kronecker-factored approximate fisher matrix for convolution layers." International Conference on Machine Learning. PMLR, 2016.

---

### Official Review · Reviewer_Xtwr · 2023-11-01

**Soundness:** 3 good
**Presentation:** 2 fair
**Contribution:** 3 good
**Rating:** 6
**Confidence:** 4

**Summary:**

This paper investigates backdoor attacks during the training process of deep neural network (DNN) and proposes a Smooth Fine-Tuning (SFT) framework to eliminate backdoors by leveraging knowledge from the Fisher Information Matrix (FIM). The research demonstrates that backdoor models tend to converge towards sharp local minima, while benign models converge towards smoother minima. Therefore, re-optimizing model parameters towards smoother minima can effectively remove backdoors. This paper introduces a novel regularizer that takes into account clean data distribution awareness and balances both model performance and backdoor purity during optimization. Additionally, ablation experiments are conducted in this study to validate the effectiveness of different components within the SFT framework.

**Strengths:**

In the realm of defense against backdoor attacks, this topic is undoubtedly intriguing. The author approaches the analysis of backdoors in DNNs from a fresh perspective, focusing on optimization. Overall, the proposed method demonstrates a noteworthy level of innovation, provides a substantial amount of detail, and the manuscript is exceptionally well-structured and well-written. In comparison to existing algorithms, the algorithm presents in this paper exhibits relatively superior performance.

**Weaknesses:**

1.In the relevant work, it has been written that the previous defense methods have high calculation costs, which limits their practicability in the actual environment. But won't the calculation of FIM in SFT increase the complexity and cost of calculation?
2.The introduction of SFT also mentions that regularized Hessian has huge calculation costs in each iteration, so that approximate methods are adopted. How to ensure that the effect can be achieved is the same?
3.Attack model on the influence of the optimization process, it does not seem to be considered.
4.The backdoor model needs to learn both clean distribution and poison distribution. This may lead to local minima or more sharp minima in the backdoor model optimization process, but does not provide a specific solution. It may be a problem to be concerned about in practical applications.

**Questions:**

1.What are the characteristics and categories of the backdoor attack methods the authors choose to compare? Does the method cover all categories of backdoor attacks?
2.Some typos and grammar errors are here:
Page 1, line 35 : wights -> weights
Page 2, line 36 : as well as -> and

---

> ### Author Response · Authors · 2023-11-16
> **Response to Reviewer Xtwr (1/2)**
>
> We thank the reviewer for the insightful comments. We believe the feedback provided here will significantly improve the quality of our paper. Please find our responses to the raised concerns as follows:
>
> ---------------------------------------------
>
> **Reviewer’s Question:** In the relevant work, it has been written that the previous defense methods have high calculation costs, which limits their practicability in the actual environment. But won't the calculation of FIM in SFT increase the complexity and cost of calculation?
>
>
> **Our Response:**
> For FIM calculation, we only need first-order differentiation (i.e., gradient) that is already computed for backpropagation. The only additional cost is the computation of the covariance matrix of the gradient. Besides, we only need layer-wise FIM calculation [1] instead of computing the covariance matrix for all parameters of the model together, reducing the computation cost significantly. To illustrate the computational efficiency of K-FAC over naive computation of FIM, let us consider an L-layered DNN model with parameters $N=\sum_{i=1}^L n_i$, where $n_i$ is the number of parameters in $i^{th}$ layer. Now, the naive approach of FIM would compute a matrix of size $N^2$ which is significantly larger than the number of elements of the matrices, $\sum_{i=1}^L n_i^2$, needs to be computed for FIM by K-FAC.
>
> We also use a faster variation of our method, f-SFT. Combining all these factors, the overall computational cost of our proposed method is significantly less as reported in Table 5 of the main paper.
>
>
> [1] Grosse, Roger, and James Martens. "A Kronecker-factored approximate fisher matrix for convolution layers." International Conference on Machine Learning. PMLR, 2016.
>
> --------------
>
>
> **Reviewer’s Question:**
> The introduction of SFT also mentions that regularized Hessian has huge calculation costs in each iteration, so that approximate methods are adopted. How to ensure that the effect can be achieved is the same?
>
> **Our Response:**
>
> Please refer to Lemma 1, which states that it is guaranteed that if we reduce the trace of FIM, the trace of Hessian must reduce too. Again, the trace of Hessian is the upper bound of the spectral norm of Hessian, implying that minimizing the trace of FIM minimizes the spectral norm of Hessian.
>
>
> ---------------------------------------------

---

> ### Author Response · Authors · 2023-11-16
> **Response to Reviewer Xtwr (2/2)**
>
> **Reviewer’s Question:** The backdoor model needs to learn both clean distribution and poison distribution. This may lead to local minima or more sharp minima in the backdoor model optimization process, but does not provide a specific solution. It may be a problem to be concerned about in practical applications.
>
> **Our Response:**
>
> In our work, we proposed a defense that purifies an already trained backdoor model that has learned both clean and poison distribution. Such defense falls under the category of test-time backdoor defense. We believe the reviewer is indicating the solutions that prevent the backdoor model from learning poison distribution (Please correct us if we are wrong). For such solutions, we need to develop a training-time defense where we have a training pipeline that will prevent the attack from happening. The training pipeline can consist of techniques such as specific augmentations like MixUp [3], where we mix both clean and poison samples to reduce the impact of the poison triggers. In recent times, several training-time defenses have been proposed such as CBD [1] and ABL [2].  Note that training-time defense is completely different from test-time defense and out of the scope of our paper. Nevertheless, we also show a comparison with these training-time defenses in Table 1.  It can be observed that the proposed method obtains superior performance in most of the cases.
>
> **Table 1: Comparison of SFT with two training-time defenses. Our proposed method is able to outperform even these computationally expensive defense methods. We consider the CIFAR10 dataset for all experiments. We consider a poison rate of 10% for all attacks. Each entry in the table is formatted as ASR/ACC.**
> |Method|Badnets  |  Blend     | Troj-one   |   Dynamic      |   WaNet  |  ISSBA|
> |:----------|:----------|:----------|:----------|:----------|----------:|----------:|
> |No Defense  |100/92.96 | 100/94.11 | 100/89.57 | 100/92.52| 98.64/92.29|99.80/92.80|
> |CBD[1]  |2.27/87.92| 2.96/89.61| **1.78**/86.18| 2.03/88.41| 4.21/87.70| 6.76/87.42|
> |ABL[2]  |3.04/87.72| 7.74/89.15| 3.53/86.36| 8.07/88.30| 8.24/86.92| 6.14/87.51|
> |SFT (Ours) |**1.86/89.32**| **0.38/92.17**| 2.64/**87.21** | **1.17/90.97**|  **2.38/89.67**|**4.24/90.18**|
>
>
> [1] Zaixi Zhang, *Backdoor Defense via Deconfounded Representation Learning*, CVPR 2023
>
> [2] Yige Li, *Anti-Backdoor Learning: Training Clean Models on Poisoned Data*, NeurIPS 2021
>
> [3] Hongyi Zhang, "mixup: Beyond Empirical Risk Minimization", ICLR 2018
>
> ------------------------------------
>
>
>
> **Reviewer’s Question:** What are the characteristics and categories of the backdoor attack methods the authors choose to compare? Does the method cover all categories of backdoor attacks?
>
> **Our Response:**
> In our work, we try to cover as many variations of backdoor attacks as possible. From a broader perspective, our defense approach could be adapted for the defense of a backdoor attack in tasks learned by supervised training. However, there may be few other variations of backdoor attacks in the literature, for instance, attacks in self-supervising learning [1], reinforcement learning [2], and Large language models [3], which we did not explore.
>
> [1] Saha, Aniruddha, et al. "Backdoor attacks on self-supervised learning." CVPR 22
>
> [2] Wang, Lun, et al. "Backdoorl: Backdoor attack against competitive reinforcement learning." arXiv preprint arXiv:2105.00579 (2021).
>
> [3] Xue, Jiaqi, et al. "TrojLLM: A Black-box Trojan Prompt Attack on Large Language Models." Thirty-seventh Conference on Neural Information Processing Systems. 2023.

---

### Official Review · Reviewer_dBn1 · 2023-11-02

**Soundness:** 3 good
**Presentation:** 3 good
**Contribution:** 3 good
**Rating:** 8
**Confidence:** 4

**Summary:**

This paper proposed a backdoor purification method based on a observation that backdoored models usually tend to converge to a bad local minima. Starting from this observation, Smooth Fine-Tuning (SFT) is proposed to erase backdoors. Besides, an efficient variant, Fast SFT is introduced to reduce the fine-tuning time.  The proposed methods are extensively evaluated on four different tasks, against 14 backdoor attacks.

**Strengths:**

1. The discovered observation is interesting, which indicates the optimization for the backdoor model training is harder and unstable than that for benign models.
2. The evaluation is extensively conducted over four different tasks.
3. It is well written and easy to follow.

**Weaknesses:**

1. Although the observation is interesting, I wondering whether the observation stands when the model size increases. This is because that increasing model complexity will make it better to achieve a tough learning goal (i.e., optimization for both clean and triggered samples), where the differences on loss surface may be not so obvious between benign model and backdoored model.

**Questions:**

Could the authors add some experiments for the effect of model capacity on the discovered observation?

---

> ### Author Response · Authors · 2023-11-16
> **Response to Reviewer dBn1**
>
> We thank the reviewer for taking the time to review our paper and providing insightful and positive comments as well as valuable feedback. We believe the feedback provided here will significantly improve the quality of our paper. We tried to address the raised concerns below.
>
>
> In Figure. 6 of Appendix A.6.1, we perform smoothness analysis for different models ranging from small to large. For example, VGG19 has around 143M parameters, whereas ResNet18 has only 11M parameters. The results show that smoothness analysis does generalize to models with different sizes. However, if the reviewer wants to see results for even larger models, we would be happy to provide that too (here and the camera-ready version).

---

### Official Review · Reviewer_DGHS · 2023-11-03

**Soundness:** 3 good
**Presentation:** 3 good
**Contribution:** 2 fair
**Rating:** 3
**Confidence:** 4

**Summary:**

This paper presents Smooth Fine-Tuning (SFT), a novel backdoor purification framework that exploits the knowledge of Fisher Information Matrix (FIM). The basic idea is to add two regularizers to the original loss to prevent the convergence to poor local minima. Some theoretical and empirical results are shown as well in the paper.

**Strengths:**

the paper is written clearly, motivation is good and results seem good (not familiar with these datasets)

**Weaknesses:**

1. Theoretical justification in Eq. 1: Thm. 1 is correct, however, it does not support the observations that backdoor attacks reach bad minima, because adding poison samples do not necessarily increase the Lipschitz constant at all! The logic from the authors is since $(L_c+L_b) \geq L_c$, the poisonous local minima have to be sharper, I guess. If so, it is definitely wrong. Otherwise, please clarify why.

2. Lack of evidence that the proposed regularized method can prevent the convergence to poor local minima: The regularizers will lead the solutions to flatter regions on the **regularized**, not original, loss landscape. This is understandable, but how to guarantee the solutions fall into smoother regions in the original loss landscape is not discussed, theoretically and empirically. I believe that this is one of the key contributions that the authors try to make. So far I do not see any evidence towards this.

**Questions:**

see my comments

---

> ### Author Response · Authors · 2023-11-16
> **Response to Reviewer DGHS (1/3)**
>
> **Response to Weakness 1**
>
> We thank the reviewer for taking the time to review our paper and providing insightful and valuable feedback. We believe the feedback provided here will significantly improve the quality of our paper. The reviewer has acknowledged the correctness of Theorem 1. However, it is suggested that more clarifications may be required to understand the implications behind Theorem 1. We aimed to address this concern below in two parts.
>
>
> **1st Part of Response to Weakness 1.**
>
> Let us consider the general training approach of a backdoor and a benign model. In general, a trained backdoor model is usually well-optimized *w.r.t.* both **clean and poison data distributions** as it is designed to perform well on both distributions. If we perform the smoothness analysis of backdoor models *w.r.t.* original training data (both clean and poison data distributions), the loss surface will be smooth. In the case of a benign model, the model is well-optimized for **clean data distribution**, hence, converges to smooth minima. Therefore, it can be observed that if the smoothness is measured *w.r.t.* the **respective training data distribution**, both models will be categorized as smooth. To clearly distinguish between benign and backdoor models in terms of their optimization characteristics, we need to consider a suitable data distribution (that represents both models well)  for smoothness analysis. In our work, we choose to conduct the analysis *w.r.t.* both clean and backdoor samples with their **corresponding ground truth labels**.
>
> Consider a training set {$\mathbf{x}, y$} = {$\mathbf{x}_c, y_c$} $\cup$ {$\mathbf{x}_b, y_b$}, where {$\mathbf{x}_c, y_c$} is the set of clean samples and {$\mathbf{x}_b, y_b$} is the set of backdoor or poison samples. If we train a model ($f$) on this dataset, the outcome would be a backdoor model, $f_b: \mathbf{x}_c \rightarrow y_c; \mathbf{x}_b \rightarrow y_b$. Now, $f_b$ should be smooth when we use training set {$\mathbf{x}, y$} = {$\mathbf{x}_c, y_c$} $\cup$ {$\mathbf{x}_b, y_b$} for smoothness analysis. Instead, if we use {$\mathbf{x}, y$} = {$\mathbf{x}_c, y_c$} $\cup$ {$\mathbf{x}_b, y'_c$}, where $y'_c$ is the original ground truth of $\mathbf{x}_b$, then $f_b$ shows non-smoothness. Now consider a scenario where we train a model ($f$) only on {$\mathbf{x}_c, y_c$}. This would give us a clean model ($f_c$). Similar to $f_b$, if we employ {$\mathbf{x}, y$} = {$\mathbf{x}_c, y_c$} $\cup$ {$\mathbf{x}_b, y'_c$} for smoothness analysis, the benign model shows smoothness. Now, there may be a query: if the model has not been trained on or seen {$\mathbf{x}_b, y'_c$}, how can the model show smoothness (loss gradient is less sensitive to trigger in $\mathbf{x}_b$)? Note that one of the most important characteristics of backdoor triggers is that they have to be stealthy (small, low intensity, not easily detectable, etc.). If triggers are not stealthy, they would be easily detectable which is not desirable. Therefore, $f_c$ treats $\mathbf{x}_b$ as a slightly perturbed version of the original clean sample ($\mathbf{x}'_c$) it was created from ($\mathbf{x}_b = \mathbf{x}'_c + \delta$). According to the stealth properties of triggers, $f_c$ should not be very sensitive to $\mathbf{x}_b$. Moreover, $f_c$ may not be sensitive to $\mathbf{x}_b$ at all depending on its robustness, which could be obtained through (commonly used) augmentation-based training. On the other hand, $f_b$ is highly sensitive to $\delta$ in $\mathbf{x}_b$ as it has been optimized to learn the trigger as well as the mapping $\mathbf{x}_b \rightarrow y_b$ almost perfectly (ASR close to 100%). Therefore, $f_b$ and $f_c$ should show different smoothness characteristics when we employ {$\mathbf{x}_b, y'_c$}. To verify how sensitive the benign and backdoor models are to the trigger, we have shown a performance comparison of them in Table 1. Higher sensitivity to the trigger should produce higher loss gradients and lower accuracy.
>
>
>
> **Table 1: Accuracy comparison (corresponding to {$\mathbf{x}_b, y'_c$}) of benign and 4 different types of backdoor models. We take 10,000 samples from CIFAR10 training set and add 4 different types of triggers without altering their ground truth labels. We then i) feed these backdoor samples to benign and backdoor models, ii) get the corresponding predictions, and iii) compare these predicated labels to the original ground truths to calculate accuracy. Lower accuracies for backdoor models indicates that we have higher loss gradients for them.**
> |Method| Badnets | TrojanNet | WaNet | ISSBA|
> |:----------|:----------|:----------|:----------|:----------|
> |Benign |90.6 |89.8|87.2|87.9|
> |Backdoor |0.00|0.00|0.41|0.26|

---

> ### Author Response · Authors · 2023-11-16
> **Response to Reviewer DGHS (2/3)**
>
> **2nd Part of Response to Weakness 1.**
>
> In Theorem 1, we also consider the {$\mathbf{x}, y$} = {$\mathbf{x}_c, y_c$} $\cup$ {$\mathbf{x}_b, y'_c$} when we discuss the Lipshitz continuity constants $L_c$ and $L_b$ of loss-gradient for backdoor and benign model. To better understand the implication of the Theorem, consider the above discussion. From the above discussion, it can be inferred that $L_b \geq L_c$ for the backdoor model as the loss gradient of backdoor model corresponding to {$\mathbf{x}_b, y'_c$} is equal to or greater than the loss gradient of the benign model. Experimentally, we observe that $L_b$ is strictly greater than $L_c$. Therefore, we can conclude that although theoretically Lipschitz constant of backdoor model  ($L_c+L_b$) is equal to or greater than the benign model $L_c$, experimentally, the total Lipschitzness of the backdoor model is strictly greater than the one for the benign model.
>
> Purifying the backdoor implies that the model will ignore any type of backdoor trigger or manipulation during testing. After purification, we often measure the performance of a model in terms of its ability to predict the original ground truth (for both clean and poison samples). From this perspective too, performing smoothness analysis *w.r.t.* the samples with their original ground truth makes sense.
>
> We have tried to provide more clarifications behind our proposed theorem here. If there are further concerns, we would be happy to address them. Thanks again for your very timely comment that helped us to elaborate this important point.

---

> ### Author Response · Authors · 2023-11-16
> **Response to Reviewer DGHS (3/3)**
>
> **Response to Weakness 2.**
>
> We thank the reviewer for raising this concern. We provide additional clarifications on why obtaining flatter regions corresponding to a regularized loss landscape would remove the backdoor.
>
>
> We would like to emphasize that the smoothness of loss landscape w.r.t. clean data distribution (i.e., training samples with their ground truth labels) is enough to remove the backdoor from the model. In other words, **achieving a flat region of loss landscape corresponding to the clean data distribution removes the backdoor** even though it is a regularized loss landscape.   *We do not necessarily need to reoptimize a backdoor model to the flat loss landscape of the corresponding benign model.* Moreover, it is highly unlikely that our proposed method will achieve the same flatter region of the original loss landscape, as the DNN optimization point is not unique **[1]**. For instance, it is highly likely that one would obtain a different optimization point for the same model optimized with the same data distribution if retrained again. Therefore, it is not important whether we are re-optimizing the backdoor model to flatter regions in the original loss landscape or not, the effect of the backdoor will be removed as long as the regularized loss landscape is flat w.r.t. clean data distribution.
>
>
> We have extensively analyzed the smoothness of the purified model for a wide range of attacks and DNN models such as ResNet34, InceptionV3, GoogleNet, DenseNet121, and MobileNetV2 reported in the “Ablation Study” section and in Appendix A.6.1. We kindly request the reviewer to check “Smoothness Analysis of SFT” in the Ablation Study (Sec. 6.3) and in Appendix A.6.1. for these results. To generate results, we applied our proposed method consisting of the regularizers during the purification phase. All these results indicate that the purified model optimized to a flatter region of loss landscape corresponding to the clean data distribution.
>
>
> [1] Chaoyue Liu et. al., "Loss landscapes and optimization in over-parameterized non-linear systems and neural networks." Applied and Computational Harmonic Analysis 59 (2022): 85-116.

---

### Author Response · Authors · 2023-11-20

We would like to thank the reviewers for their time and effort in reviewing our paper and providing valuable feedback. We appreciate that the reviewers find our paper is well-written and well-motivated (DGHS, dBn1, Xtwr, cxbT), a novel perspective of backdoor analysis and a novel method (dBn1, Xtwr, cxbT), and extensive experiments (dBn1), etc.

We are grateful to the reviewers for their constructive feedback that helped us to improve the paper. We have tried to address each concern raised by the reviewer in our response. Specifically, following reviewer DGHS's comments, we have provided further clarification regarding the Lipschitzness of the loss gradient of a backdoor model and the study of smoothness properties corresponding to the clean data distribution. We have also explained how the backdoor removal process is related to the regularized loss landscape. Moreover, we have addressed all comments from the reviewers dBn1, Xtwr, cxbT in the rebuttal and revised the paper accordingly, especially the Appendix.

We hope our response will address the reviewer’s comments and suggestions. However, as the discussion period will end soon, we would highly appreciate reviewers' further feedback (if any) and would be happy to address any further comments.

---

### Public Comment · ~Zhiwei_Jia1 · 2023-11-23
**Related Work**

Hi authors. I like your work and think it would be great to add the following work [1], which also exploits the knowledge of the FIM for improving generalization (instead of backdoor defense, which is very relevant), to the related work. Thanks.

[1] Information-Theoretic Local Minima Characterization and Regularization

---

### Meta-Review · Area_Chair_8KRH · 2023-12-10

**Metareview:**

This work studied the post-processing backdoor purification by encouraging smoothness.
It provided theoretic analysis and empirical studies to demonstrate that backdoor model locates at a shaper local minima than the benign model in the loss landscape. Extensive experiments are provided.

There are detailed but diverse reviews. The most critical point is the correctness of theorem 1 and its implication that backdoor model locates at a shaper local minima than the benign model in the loss landscape. Reviewer DGHS proposed two insightful comments. The authors responded a lot, but didn’t directly and clearly clarify the question. A larger upper bound of the smoothness, i.e., the larger Lipschitz constant, doesn’t mean sharper minima. If it was lower bound, then one could claim.
Besides, theorem 1 is trivial, but the proof in appendix is questionable. For example, the first step in Eq. (12) is not a rigorous derivation.

Moreover, all analysis is based on the hessian matrix of all parameters, but the actual computation is the layer-wise hessian matrix, which is mentioned in the rebuttal. There is likely huge gap between global and layer-wise hessian matrix, leading to totally different conclusion between theoretical and real value on smoothness.

The rich experiments on different applications are appreciated. The smoothness perspective is also interesting, although there are also some attempts in previous works. However, the behind theoretical analysis and claim is questionable, and it may misguide the research community of backdoor learning, which outweighs the contribution of this work.

**Justification For Why Not Higher Score:**

see above

**Justification For Why Not Lower Score:**

n/a

---

### Decision · Program_Chairs · 2024-01-16

Reject